# Emulating Cued Recall of Abstract Concepts via Regulated Activation Networks

**Rahul Sharma** [†,‡] , **Bernardete Ribeiro** [‡] , **Alexandre Miguel Pinto** [‡] and **Amílcar Cardoso** [*,‡]

Center of Informatics and Systems (CISUC), DEI, Polo-II University of Coimbra, Pinhal de Marrocos,
3030-290 Coimbra, Portugal; rahul@dei.uc.pt (R.S.); bribeiro@dei.uc.pt (B.R.); ampinto@dei.uc.pt (A.M.P.)

* Correspondence: amilcar@dei.uc.pt
† Current address: Centre for Informatics and Systems of the University of Coimbra Polo II, Pinhal de Marrocos,
3030-290 Coimbra, Portugal.
‡ Dr. Rahul Sharma is the main contributor under the supervision of Dr. Bernardete Ribeiro, Dr. Alexandre
Miguel Pinto, and Dr. Amílcar F. Cardoso.

**Abstract:** Abstract concepts play a vital role in decision-making or recall operations because the associations among them are essential for contextual processing. Abstract concepts are complex and difficult to represent (conceptually, formally, or computationally), leading to difficulties in their comprehension and recall. This contribution reports the computational simulation of the cued recall of abstract concepts by exploiting their learned associations. The cued recall operation is realized via a novel geometric back-propagation algorithm that emulates the recall of abstract concepts learned through regulated activation network (RAN) modeling. During recall operation, another algorithm uniquely regulates the activation of concepts (nodes) by injecting excitatory, neutral, and inhibitory signals to other concepts of the same level. A Toy-data problem is considered to illustrate the RAN modeling and recall procedure. The results display how regulation enables contextual awareness among abstract nodes during the recall process. The MNIST dataset is used to show how recall operations retrieve intuitive and non-intuitive blends of abstract nodes. We show that every recall process converges to an optimal image. With more cues, better images are recalled, and every intermediate image obtained during the recall iterations corresponds to the varying cognitive states of the recognition procedure.

**Keywords:** computational psychology; computational cognitive modeling; machine learning; concept blending; conceptual combinations; recall; computational creativity

## 1. Introduction

Concepts are an important object of research in cognitive and psychological research. Usually, the conceptual representations are process-oriented, symbolic or distributed, and knowledge-based [1–3]. In general, a hierarchical structure defines an organization of concepts where the concrete concepts are placed in the lower level, and the abstract Concepts occupy the higher levels (see the example of the vehicle and cars in Figure 1). Therefore, abstract concepts are also seen as the generalization of concrete concepts [4,5]. Abstract concepts are studied mathematically [6] and theoretically [7,8], but computational studies are scarce [1]. This article uses a computational model, regulated activation network (RAN) [9–11], capable of building representation of convex abstract concepts, which are later used in recall simulations.

The prime aspect of this article is to emulate the recall procedure, that can be viewed as the cognitive process of remembering. Context plays an eminent role in the recall of concrete concepts (such as the word "table"), which is often termed as concreteness effect. The concreteness effect is expressed through the dual-coding theory [12,13] and the context availability hypothesis [14]. According to this theory, individually, abstract concepts are abstruse in context retrieval when compared with concrete concepts; therefore, their

recall procedure is complex. However, an interesting work suggests that context retrieval of abstract concepts is possible when response pairs of abstract concepts are related to one another, thus providing context for the abstract stimuli [15]. In RAN modeling we can learn associations among concepts (including the abstract concepts). These learned associations provide adequate context relations among the abstract concepts. In this work, we exploit these learned associations among the abstract concepts to simulate a regulated recall operation.

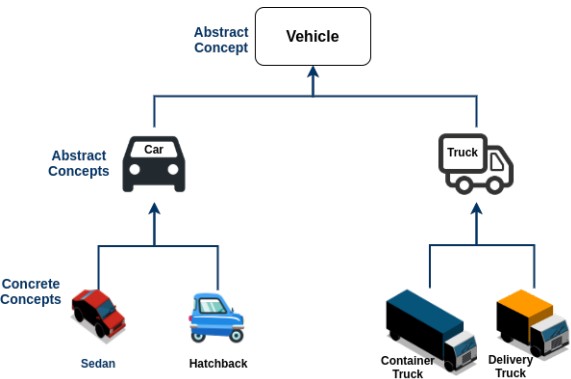

**Figure 1.** Hierarchy abstraction of concrete and abstract concepts.

In recent years, advances in technologies have played an essential role in cognitive and psychological research, e.g., use of devices like GP3 [16], TheEyeTribe [17], and electroencephalography (EEG) [18,19] to study visual attention. In this article, we focus on representations of computational models that are also very useful in understanding the psychological and cognitive phenomena, validating the existing cognitive theories, and helping to formulate fresh ideas related to cognition [20–23]. The representations of these computational approaches are either symbolic (amodal), distributed (multimodal), or hybrid [24], which helps in simulating or understanding various cognitive phenomena. Below are a few examples of computational models or architecture used to study psychological and cognitive phenomena based upon their representation:

- Symbolic: Adaptive Control of Thought–Rational (ACT-R) [25] is a symbolic architecture intended to model memory [26] and simulate attention [25,27], decision-making [28], recognition [29], and forgetting [29].
- Distributed: Multimodal approaches such as artificial neural networks (ANNs), including the restricted Boltzmann machine (RBM) [30], deep neural networks [31], stacked auto-encoders [32], and convolution neural networks (CNN) [33], have significant contributions in feature recognition [34] and distributed memory representation [35]. Methods like Random Forests have also been used in studies related to visual attention [36].
- Hybrid: Cognitive architectures like Connectionist Learning with Adaptive Rule Induction On-line (CLARION) [37] simulate scenarios related to cognitive and social psychology.

This article makes use of the RAN's hybrid nature and modeling to build the representation of convex abstract concepts and further simulate recall of abstract concepts. F First, the model generation takes place with four basic steps of the RAN approach [9], i.e., concept identification, concept creation, inter-layer learning, and upward activation propagation. An intra-layer procedure also takes place at all the layers to identify the association among the concepts at the same level. Further, these learned associations are uniquely interpreted to determine whether the impact of their learned weights is inhibitory, excitatory, or neutral. Later, these impacts are applied to obtain a regulatory effect on peer concepts (abstract concepts or input layer concepts) during recall operation. A Toy-data problem was used for modeling with RAN and demonstrating the novel geometric back-propagation algorithm for the simulation-cued recall operation. The benchmark dataset of

the image domain, MNIST, is also used to demonstrate the cued recall experiment. These experiments also show how blends of abstract concepts can be recalled. To summarize, the following are the main contributions of the article: first, the impact factor calculation to determine the inhibitory, excitatory, or neutral effect of one node over other; second, the novel intra-layer regulation algorithm for the use of the impact factor in order to regulate the activation of other concepts; third, the novel geometric back-propagation algorithm and recall simulations using the geometric back-propagation algorithm.

The remainder of this article is organized in the following way: Section 2 puts forward the state of the art related to recall operations; RAN modeling, the intra-layer regulation algorithm, and the geometric back-propagation algorithm are detailed in Section 3 using a Toy-data problem; the cued recall demonstration with the MNIST dataset is reported in Section 4; Section 5 concludes the article.

## 2. Related Work

Recall or retrieval is a cognitive process [38] of remembering a thing or an event. While recalling, the brain activates a neural assembly that was created when the original event occurred [38]. In psychology, there is a plethora of articles studying the recall process. Psychologists used free-recall, cued-recall, and serial-recall as tools to investigate memory processes [39]. Recall has been used to study the effect of cognitive strategies, such as chunking and the use of mnemonics for memorization of things (such as large numbers) [40]. One interesting study reported the benefits of subsequent recall in retrieval operations where memories are related or competing [41]. The proverb "practice makes a man perfect" relates to the fortification of memory, and an investigation shows how retrieval plays an important role in this memory strengthening [42]. Technologies such as functional magnetic resonance imaging (fMRI), magnetic resonance imaging (MRI), positron emission tomography (PET), and electroencephalography (EEG) played an active role in validating many recall related hypotheses [41,43–45].

Notable contributions to the modeling of memory recall procedures are observed. Based on the temporal context model [46,47] of human behavior, human memory performance was modeled using a probabilistic approach during free-recall experiments [48]. A computational model of interaction between the prefrontal cortex and medial temporal lobe in memory usage was designed to study the prefrontal control in a recall process [49]. The model was a simple neural network with quick and flexible reinforcement learning exhibiting strategic recall. Another computational model differentiates recall from the recognition process depending upon the number of cues involved in the retrieval procedure [50]. For encoding, the model used an inference-based model of memory [51], and retrieval was carried out using a Bayesian observer model [52]. A large number of computational psychology contributions examining the recall process and recognition using the neural networks are available [53–56].

An interesting study simulated the free-recall process using the ACT-R architecture [57], showing that the classical effect of primacy and recency can be recreated through rehearsal theory based upon ACT-R and Baddeley's phonological Loop [58]. ACT-R architecture was also used to propose a new theory of memory retrieval to predict for intricate serial and free-recall operations [59]. This research also focused on the prospects of associative learning by introducing a strengthening and decaying mechanism depending upon the similarity of the input stimulus. The serial recall has been modeled in a scientific contribution using ACT-R architecture to explain the processes involved while recalling a list of words [60]. The traditional ACT-R recall operations had a limitation: here, the memory access depends upon limiting the capacity of the activation process, consequently inducing errors in the contents being recalled. This theory overcomes the limitations by predicting the latency and errors in a serial recall process.

The free recall process was also modeled using CLARION to determine the role of distractions in an incubation task [61]. This study made a striking observation that rehearsals play an important role in memory consolidation during the free recall procedure,

and distractions can hinder the free recall and eventually effect memory strengthening. CLARION was also used to emulate, acquire, and expound human-centric data relevant to incubation and insight through free recall, lexical decision, and problem-solving tasks [62].

This article introduces a novel algorithm named geometric back-propagation, which enable us to simulate the recall simulation using RAN modeling. The main objective of the experiments in this paper is to demonstrate the role cue (activation) on abstract concepts (nodes) in recall operations. An additional goal is to show that when larger number abstract concepts participate (i.e., more cues are available) in recall operations, then better recall is observed in the experiment.

## 3. RAN Methodology to Simulate Recall Operations

Here, we describe the emulation of the recall process using RAN modeling. For background understanding in Section 3.2 we describe RAN modeling along with two learning mechanisms, i.e., inter-layer and intra-layer learning. Having explained RAN modeling, the two contributions of this article are elucidated: first, the regulation mechanism is described, the biological inspiration behind this operation is desibed in Section 3.1. Secondly, a novel geometric back-propagation algorithm is proposed that propagates activations from abstract level to input Level. RAN methodology and the article's contributions are illustrated using a Toy-data set. At the end of this section, the experiments of RAN modeling with the Toy-data are also reported, demonstrating the recall operation.

### 3.1. Biological Inspiration of Regulation Operation on RAN's Modeling

The nerve cell (neuron) consists of several main components: the dendrites, the cell body, and the axon, as shown in Figure 2. When an electric signal traverses the whole axon and reaches one of its terminations, it releases chemicals called neurotransmitters, which diffuse across the synaptic gap and are absorbed by the receptive neuron's dendrite. Depending on the neurotransmitter, this absorption can either enhance or inhibit the receptive neuron's activation.

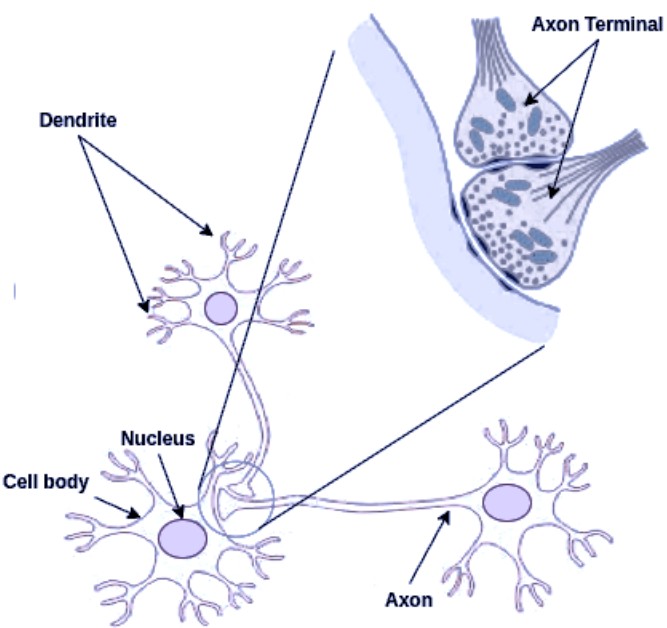

**Figure 2.** Biological neuron showing the axoaxonic synapses.

Other kinds of synapses occur in biological brains, such as axoaxonic synapses, as shown in Figure 2. These synapses occur when the axon of a neuron connects to the axon of another neuron instead of to its dendrites. Such configuration usually plays a regulatory role by mediating presynaptic inhibition and presynaptic facilitation [63]. By virtue of artificial axoaxonic synapses, this contribution realizes the inhibitory, excitatory,

and neutral activation propagation phenomenon in RAN modeling, which is used to induce a regulatory effect on the activation at nodes during recall operation.

### 3.2. Abstract Concept Modeling with RAN

This section is dedicated to describing convex abstract concept modeling with RAN [11]. To demonstrate RAN methodology, a Toy-data problem is used—see Figure 3. The Toy-data are synthetically produced by generating a 2D dataset with five classes. In Figure 3, we can see that out of the five clusters, three are far apart from one another; however, two clusters are very close to each other. This arrangement of clusters was introduced into the Toy-data problem to demonstrate the excitatory and inhibitory impact of concepts, representing each cluster at an abstract level. The dataset consists of 1800 data instances with an equal distribution in all of the classes. RAN modeling is performed using the four basic steps, where step 1 and step 4 consist of two concurrent operations.

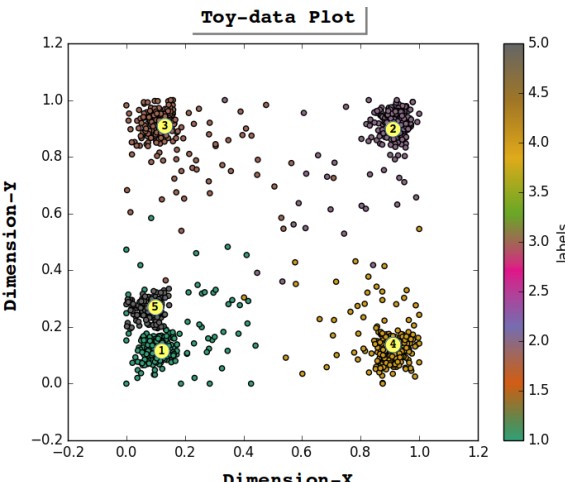

**Figure 3.** Graph of 2D Toy-data with five clusters, along with their respective cluster centers (1, ..., 5).

### 3.2.1. Step 1a: Convex Concept Identification (CCI)

CCI is a method to determine convex groups in a given dataset. In RAN, each data instance is considered a point in n-dimensional geometric feature space, inspired by the theory of conceptual spaces [64]. In this method, we also determine the cluster centers that are used in the inter-Layer learning operation (see Section 3.2.4). Step 1a in Figure 3 shows input Layer 0; here, nodes $S_1$ and $S_2$ correspond to the dimensions of the input Toy-data. To identify the five convex groups, the K-mean [65] clustering algorithm is chosen as the concept identifier (CI), and the value of $K$ is set to 5 to determine five clusters. Five cluster centers are also identified in this process, as shown in Figure 3, as cluster representative data points (CRPD). Any clustering algorithm can act as a concept identifier provided that it enables the identification of convex regions along with their respective cluster centers.

### 3.2.2. Step 1b and Step 4b: Concept Similarity Relation Learning (CSRL)

CSRL is an intra-layer operation in RAN modeling explained in [66]. The main purpose of this process is to determine the alikeness among the concepts and associate them through a similarity relation. This relation also simulates the behavior of activation found in biological neurons; i.e., affine neurons are activated concurrently upon receiving input stimuli, whereas unrelated neurons remain relatively inactive for the same input stimulus. This phenomenon is expressed mathematically through Equation (1) to calculate a pair-wise relation/weight $w_{m \rightarrow n}$ between node $m$, and node $n$ at a layer. The numerator $(1 - |A_m^I - A_n^I|)$ calculates the similarity of activation ($A_m^I$ is the activation of $I$th instance of propagated data at node $C_m$ in a layer, and similarly, $A_n^I$ is the activation of $I$th instance of propagated data at node $C_n$. $m \neq n$ and $m, n$ are integers.) of node $m$ w.r.t. node $n$, and the product $(1 - A_m^I) * (1 - A_n^I)$ is used to reduce the impact of similarity on weight $w_{m \rightarrow n}$

when both activations (i.e., $A_m^I$, and $A_n^I$) are very close to 0, though similar. Consequently, we obtain a symmetric $k \times k$ matrix as learned concept similarity relation weights (CSRW) among the nodes within the layer.

$$w_{m \to n} = \frac{\sum_I [(1 - |A_m^I - A_n^I|) - (1 - A_m^I) * (1 - A_n^I)]}{\sum_I [1 - (1 - A_m^I) * (1 - A_n^I)]} \tag{1}$$

where $m \in 1, ...., k; n \in 1, ...., k;$ and $m \neq n$.

This learning mechanism is performed two times while modeling with Toy-data: First, in Step 1b at the input layer 0, (see Figure 4, step 1b), and second, when the input data is propagated upward to the convex concept layer 1. The learning at layer 0 has a size of $2 \times 2$, as the input layer has two nodes. In contrast, the learning at layer 1 has a size of $5 \times 5$ (see Figure 5 for the CSRL weights).

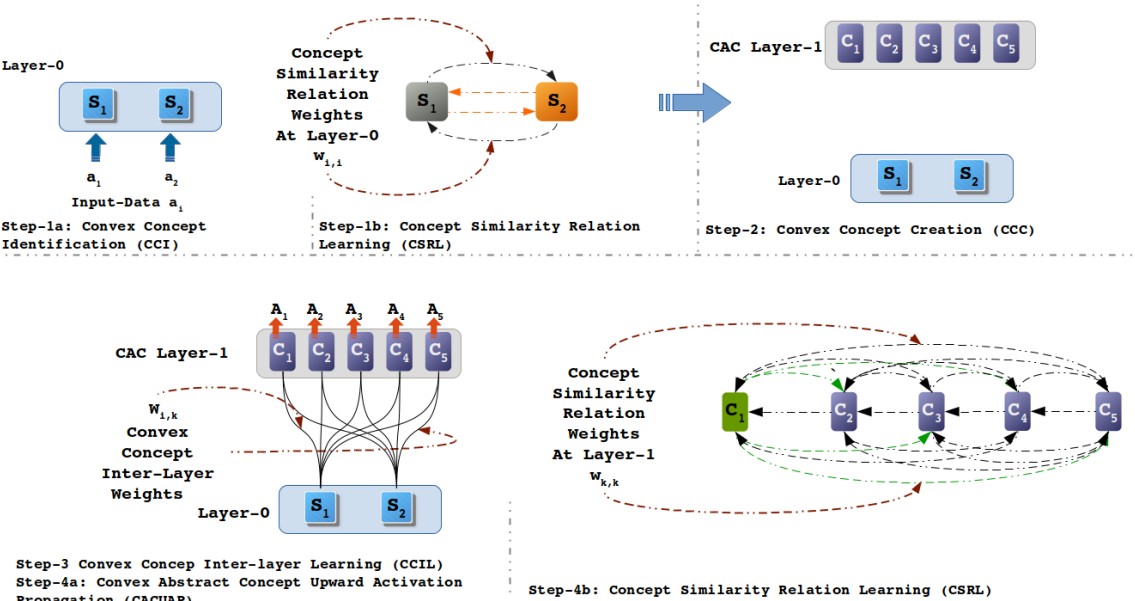

**Figure 4.** Regulated activation network (RAN) convex abstract concept modeling process. The procedure displays the four steps in RAN modeling. This figure shows the three learning procedures, i.e., two similarity relation learning procedures at two layers, and one inter-layer learning procedure between layer 1 and layer 0. In step 1, similarity relation learning (step 1a) is performed along with the concept identification process (step 1b). Similarly, in step 4, similarity relation learning (step 4b) is performed together with the upward activation propagation method (step 4a).

### 3.2.3. Step 2: Convex Abstract Concept Creation (CACC)

Convex abstract concept creation is a method of dynamically creating a layer that consists of nodes as an abstract representative of categories determined in the convex concept identification process (see Section 3.2.1). For instance, step 2 in Figure 4 depicts the creation of new a convex abstract concept (CAC) layer 1 with five nodes ($C_1$, ..., $C_5$). These five nodes ($C_1$, $C_2$, $C_3$, $C_4$, and $C_5$) represent clusters 3, 1, 4, 2 and 5 in Figure 3, respectively. The count of the CAC layer nodes depends upon the number of clusters identified in the CCI operation at the input layer; i.e., if $k$ clusters were determined in the CCI mechanism, then in step 2, a new layer is created consisting of $k$ nodes.

### 3.2.4. Step 3: Convex Concept Inter-Layer Learning (CCILL)

Besides intra-Layer learning in Steps 1b and 4b, the second learning mechanism in RAN modeling is used to identify the association among the nodes at the CAC layer and input layer. Since the nodes are the abstract representative of the clusters identified in the CCI process, and according to the theory of prototype, the cluster center (i.e., the CRPDs)

is the most probable representative of a cluster [67–69]. Therefore, the cluster centers are understood as an association among the CAC layer nodes and input layer nodes, as depicted by Equation (2).

$$W_{k,i} = \begin{bmatrix} W_{1,1}, W_{1,2}, \ldots, W_{1,n_a} \\ \cdots \\ W_{t,1}, W_{t,2}, \ldots, W_{t,n_a} \\ \cdots \\ W_{n_A,1}, W_{n_A,2}, \ldots, W_{n_A,n_a} \end{bmatrix} = \begin{bmatrix} C_1 \\ \cdots \\ C_t \\ \cdots \\ C_{n_A} \end{bmatrix} \quad (2)$$

where $k = 1, 2, \ldots, n_A$, and $i = 1, 2, \ldots, n_a$

In Equation (2) we can see that the coordinates (feature values of data) of centers identified in the CCI process are assigned as learning $W$, i.e., convex concepts inter-layer wiights (CCILW). In the experiment with Toy-data, we learned a $5 \times 2$ weight matrix between the five nodes at CAL layer 1 and two nodes of input layer 0, as shown in Figure 4, step 3. Having completed step 3, a basic RAN model is obtained consisting of input layer 0, CAC layer 1, learning between two layers, and learning among the nodes at input layer 0.

**Figure 5.** The concept similarity relation learning (CSRL) weight matrices learned with Toy-data and their corresponding impact dactor ($\sigma$) at layer 0 and layer 1. $\sigma$ is calculated using Equation (5).

### 3.2.5. Step 4a: Convex Abstract Concept Upward Activation Propagation (CACUAP)

This step is used to propagate i-dimensional input data vector $a_i$ to the CAC layer and to obtain k-dimensional data vector $A_k$. This mechanism is used in two stages: in the first stage, Euclidean distance is calculated among the input data $a_i$ and all the CCILWs $W_{k,i}$. This distance is further normalized (using the denominator in Equation (3)) to obtain distance in the range [0, 1](in RAN's modeling, the activation values of are, by definition, real values in the [0, 1] interval—and in such a setting, in an $n$-dimensional space, the maximum possible Euclidean distance between any two points is $\sqrt{\sum_{i=1}^{n}(a_i - 0)^2} = \sqrt{n}$, where $a_i = 1$.).

$$d_k = \frac{\sqrt{\sum_{i=1}^{n_a}(W_{k,i} - a_i)^2}}{\sqrt{n_a}} \quad (3)$$

In the second stage, the normalized distance obtained from Equation (3) is transformed non-linearly, establishing a similarity relation conforming to the following three conditions: (1) $f(d = 0) = 1$, i.e., when distance is 0, similarity is 100%; (2) $f(d = 1) = 0$ i.e., when distance is 1, similarity is 0%; and (3) $f(d = x)$ is continuous, monotonous, and differentiable in the [0, 1] interval.

$$f(x) = (1 - \sqrt[3]{x})^2 \quad (4)$$

This similarity relation equation (Equation (4)) transforms the distance values observed at each node in the CAC layer into its similarity value. These similarity values act as degree

of confidence (DoC) values to recognize the category being represented by the nodes at the CAC layer. Upon propagating all input values to the CAC Layer, the observed outputs $A_k$ are used to perform concept similarity relation learning (CSRL), as shown in Figure 4, step 4b. After completing step 4b, the RAN modeling procedure terminates, and a model is obtained, as shown in step 3 in Figure 4.

In order to build more than one layer, all of the steps are repeated iteratively, and the output of all the intermediate CAC layers is pipelined as input to the new layer being built.

### 3.3. Regulation Mechanism

The regulation operation in RAN modeling is performed in three steps: first, an impact factor of the CSRL matrix is deduced; second, the intra-layer (IL) contribution of activation at a node by another node in the same layer is determined; third, activation at a node by a function of self-activation and intra-layer activation induced by other nodes on the latter are obtained.

### 3.3.1. Impact Factor ($\sigma$) Construction and Interpretation

The impact factor is a function that interprets the CSRL weight values (in the range [0, 1]) as excitatory, inhibitory, or neutral weights. The purpose of CSRL weights is to determine how concurrently two nodes (e.g., $S_1$ and $S_2$) are active. If the CSRL weight is intermediate, i.e., 0.5, it signifies that the two nodes are 50% concurrently active (depicting a state of confusion). Therefore, these nodes do not have an impact on each other in the same layer. If the CSRL weight of the two nodes were "0", then the two nodes were never active simultaneously. This also indicates that the two nodes are inhibitors of each other. Finally, if the CSRL weights of the two nodes were "1", then the nodes were always active conjointly, exciting the activation of each other.

$$\sigma_{m \to n} = [2 * (W_{m \to n} - 0.5)]^3 \tag{5}$$

The aforementioned comprehension of CSRL weights $W$ is exhibited by a mathematical Equation (5), where $\sigma_{m \to n}$ is the impact of node $m$ over node $n$. Figure 6 shows a graphical view of the impact factor $\sigma$ (Equation (5)), depicting the excitatory, inhibitory, or neutral interpretations of CSRL weights. Figure 5 shows the CSRL and their respective $\sigma$ weights for both layers. At layer 0, the nodes $S_1$ and $S_2$ have a very minimal excitatory impact on each other (in Figure 3, every node at layer 1 can be related to clusters $C_1$, ..., $C_5$ serially). However, at layer 1, node $C_1$ has no impact on node $C_2$ (and vice versa). There are many negative weights in the $\sigma$ matrix of layer 1, indicating that these nodes inhibit each other. In Figure 3, we see that the clusters $C_2$ and $C_5$ are very close, and the activations observed at both of the nodes must be very similar. Hence, high CSRL weight is learned between nodes $C_2$ and $C_5$. Notably, both exhibit good excitatory behavior towards each other.

### 3.3.2. Intra-Layer Activation

The objective of calculating intra-layer (IL) activation is to determine the amount of activation a node $n$ receives from all the other $m$ nodes of the same Layer. To obtain the intra-layer activation at node $n$, the approach must address three prospects. First, intra-layer activation must consider the impact ($\sigma$) of excitatory, inhibitory, or neutral effects of all $m$ nodes over node $n$. Second, the current activation of $m$ nodes and their CSRL weight ($Wm \to n$) to node $n$ should be considered in calculating the activation of node $n$. Third, the intra-layer activation computed for node $n$ must be in the range [0, 1]. Equation (8) conforms to all three requirements.

$$\chi m = (a_m * W_{m \to n}) \tag{6}$$

$$\neg \chi m = (1 - a_m) * (1 - W_{m \to n}) \tag{7}$$

$$IL(a_n) = \frac{\sum_m \sigma_{m \to n}(\chi_m + \neg\chi_m)}{\sum_m \sigma_{m \to n}} \tag{8}$$

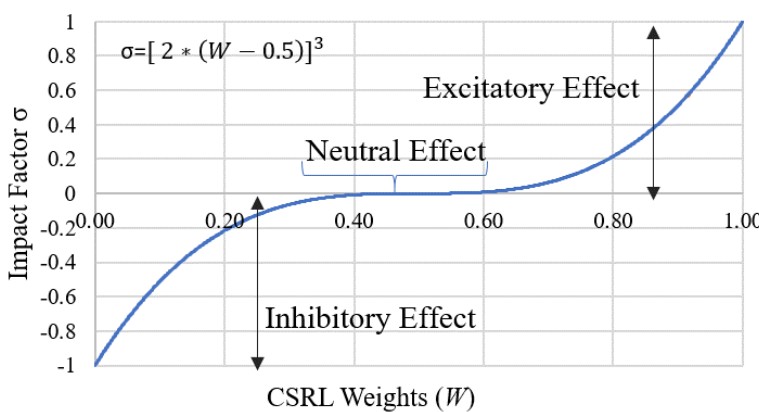

**Figure 6.** Excitatory, inhibitory, and neutral effect of CSRL weights (W) when transformed using the impact factor $\sigma$.

### 3.3.3. Intra-Layer Regulation

To identify the actual activation AA($a_n$) at node $n$, this operation uses a regulation factor ($\rho$) to decide the share of self-contribution of activation by node $n$ and intra-layer activation at node $n$, i.e., IL($a_n$). Equation (9) shows the mathematical function for the intra-layer regulation operation. From Equation (9), we can observe if the $\rho$ is '0'; i.e., without any regulation, only the activation of node $n$ contributes to the actual activation.

$$AA(a_n) = (1 - \rho) * a_n + \rho * IL(a_n) \tag{9}$$

Algorithm 1 presents the intra-layer regulation operation in an algorithmic form. This regulation operation has its importance when propagating the activation from an abstract concept layer to the input layer, as described in Section 3.4.

---

**Algorithm 1:** Intra-Layer Regulation

---

**Input:** current activation $a_n$ at node $n$ at layer L.
**Input:** CSRL W at layer L
**Input:** impact matrix $\sigma$ at layer L
*Initialization*: regulation factor $\rho$, between [0, 1];
**foreach** $a_n$ in L **do**
⎸Calculate IL($a_n$), using Equation (8);
⎸Calculate actual activation AA($a_n$), using Equation (9);
**end**
**return** AA($a_n$)

---

### 3.4. Geometric Back-Propagation Operation

Geometric back propagation (GBP) is a downward propagation mechanism in RAN modeling. This method enables us to determine an activation vector $a_m$ ($< a_1, .., a_i, .., a_m >$) at layer *L-1*, for an expected activation (E-A) vector $A'_n$ ($< A'_1, .., A'_j, .., A'_n >$) at layer *L*. This operation is a window operation that takes place between two adjacent layers, i.e., layers *L* and *L-1*. For instance, if the RAN model has three layers, $L_0$ (input layer), $L_1$, and $L_2$ (output layer), then two GBP operations take place, first between $L_2$ and $L_1$, and then between $L_1$ and $L_0$. Figure 7 shows the single-window operation between two layers.

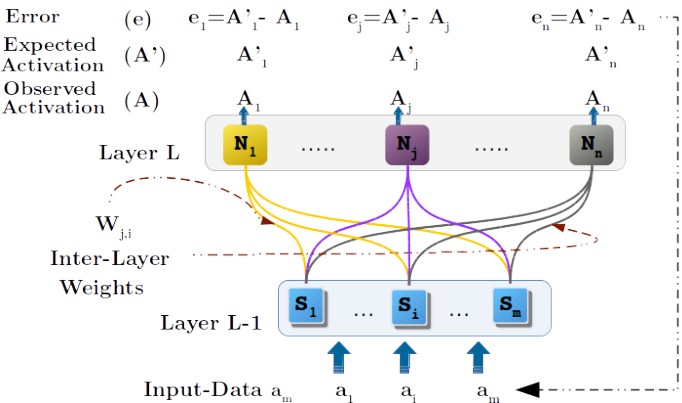

**Figure 7.** Single-window operation in geometric back-propagation operation. The figure also shows the error calculation and propagation.

The GBP mechanism commences with an expected activation ($A'_n$) vector at layer *L*. Next, a starting input vector $a_m$ ($< a_1, .., a_i, .., a_m >$) is injected in layer *L-1*. Now, we enter into a cycle where we propagate the activation of the nodes in layer *L-1* upwards to layer *L* and determine the observed activation (O-A) vector $A_n$ ($< A_1, .., A_j, .., A_n >$) at layer *L*. Furthermore, an error vector *e* is calculated using $A'_n$ and $A_n$ (expected and observed activation vector) through Equation (10). The error vector *e* is used to determine an accumulated delta value $\triangle_{a_i}$ (see Equation (12)) based upon the function expressed by Equation (11). This $\triangle_{a_i}$ value is then added to the activation vector at layer *L-1* (see Equation (13)) to obtain a new input vector $a_m^{new}$. Additionally, the cycle is repeated with the new $a_m^{new}$ input at layer *L-1* until the error is minimized or the cycle equals the user-defined maximum iteration threshold. Algorithm 2 presents the detailed geometric back-propagation algorithm. As mentioned earlier, the GBP operation takes place between two consecutive layers. However, if the hierarchy has more than two layers, then with the window operation, it is possible to propagate down the injected E-As at the nodes of the top-most layer *L*, to input layer 0.

$$e_j = A'_j - A_j \tag{10}$$

$$\triangle_{a_i, A_j} = (W_{j,i} - a_i) * (e_j) \tag{11}$$

$$\triangle_{a_i} = \sum_{j=1,..,i} \triangle_{a_i, A_j} \tag{12}$$

$$a_i^{new} = a_i + \triangle_{a_i} \tag{13}$$

### 3.5. Recall Demonstration with Toy-Data

There are two types of experiments performed in this section: first, the single-cue recall (SCR) operation, where the recall is performed based upon the expected activation by one node in an abstract concept layer and, second, the multiple-cue recall (MCR) mechanism, where the recall procedure is carried out for the expected activation at all the nodes in the abstract concept layer. The experiments demonstrated in this section use the two-layered model generated with RAN methodology (see the two-layered model obtained at step 3 in Figure 4 in Section 3.2). In the generated RAN model, the five abstract concept nodes ($C_1, C_2, C_3, C_4$, and $C_5$) correspond to clusters 3, 1, 4, 2 and 5, respectively (see Figure 3 for the clusters). In both SCR and MCR operations, the five abstract concept nodes at layer 1 are injected with an expected activation set. Furthermore, the geometric back-propagation algorithm (Algorithm 2) performs one thousand iterations of downward propagation of activation to obtain appropriate values at input layer 0 as recalled activation. In all of the recall simulations, the GBP operation is initialized with values 0 and 0.60 as *Starting-Point*, and *maxIter* is set for 1000 iterations. The expected activation varies with the experiment, and two sets of the regulation factor were determined empirically to demonstrate both recall procedures.

3.5.1. Single-Cue Recall (SCR) Experiment

In SCR experiments, the objective was to determine the recalled activation at input layer 0 by injecting binary activation values as expected activation in abstract concept layer 1. The expected activation vector contains value 1 for only one abstract concept node, and for the remaining nodes, 0 is assigned. In all SCR experiments, six regulation factors $\rho$ (0%, 0.5%, 0.75%, 1%, 1.25% and 1.5%) were used. Table 1 logs the E-A and the thousandth iteration value of O-A for the five SCR experiments. The following are the five experiments to demonstrate the SCR operation along with observations:

- **Exp-1:** This is the first experiment in which we injected an E-A vector *[0, 0, 0, 0, 1]* of activation at abstract concept layer 1. The objective was to recall activations at input layer 0 for which a very high activation was observed at node $C_5$ at layer 1 and comparatively lower activation for the other four nodes at layer 1. The GBP algorithm was executed six times with an E-A of *[0, 0, 0, 0, 1]* for the six different regulation factors ($\rho$). The observation for Exp-1 (see Table 1) shows that with a $\rho$ of 0.75%, the maximum activation of 0.85 was observed at node $C_5$. As expected, the node $C_2$ received good activation because nodes $C_2$ and $C_5$ represent clusters 1 and 5 (see Figure 3), which are close to one another. Figure 8a shows the six trajectories for the six regulation factors; each trajectory is formed by one thousand iterations. In Figure 8a, the yellow marker shows the CRDP of cluster $C_5$ and the trajectory with $\rho$ of 0.75% converge closest to this CRPD. Thus, an activation vector *[0.1, 0.24]* is recalled at input nodes $[S_1, S_2]$ for the given E-A vector *[0, 0, 0, 0, 1]*.
- **Exp-2:** In this experiment, the E-A provided to the GBP algorithm was *[0, 0, 0, 1, 0]* to recall activation at layer 0, which is strongly represented by node $C_4$. For each regulation factor, the GBP algorithm was run; the O-As obtained at layer 1 are listed in Table 1, and the corresponding recalled activation at layer 0 is shown in Figure 8b. From the observations, it can be deduced that the experiment with $\rho$ of 0.75% produced the best outcome and recall activation *[0.9, 0.9]* for input layer 0.
- **Exp-3:** In this experiment, the GBP algorithm was supplied with an E-A vector of *[0, 0, 1, 0, 0]* to recall the input layer 0 vector, which is represented by node $C_3$ at layer 1. Figure 8c and Table 1 shows the recall trajectories at layer 0 and the O-A vector at layer 1, respectively. The experiment with the regulation of 0.75% displayed the best representation. A vector *[0.92, 0.11]* was recalled at layer 0 for the injected E-A vector.
- **Exp-4:** The aim of this experiment was to recall an input vector that closely represents the abstract concept node $C_2$ by feeding the GBP algorithm with an E-A vector of *[0, 1, 0, 0, 0]*. After applying the six regulation factors to each GBP operation, it was observed that the experiment with $\rho$ of 0.75% displayed the best result. Table 1 shows the O-A for the E-A. Figure 8d shows the trajectories of the recalled values and shows the best outcome with $\rho$ of 0.75% that converges to an activation vector *[0.14, 0.07]*.
- **Exp-5:** This experiment shows the recall vector obtained by initializing an E-A vector *[1, 0, 0, 0, 0]* with all GBP experiments with the six regulation values. Unlike the previous experiments, the best O-A was obtained with a regulation factor of 1%. Figure 8d shows the recall outcome for all six regulation factors. At input layer 0, the recall operation with $\rho$ of 1% results into an activation vector *[0.15, 0.91]* for the given E-A.

---

**Algorithm 2:** Geometric Back-Propagation Operation

---

**Input:** A *ExpectActivation* activation $A'_n$ ($< A'_1, .., A'_j, .., A'_n >$) at layer *L*, with *n* nodes

**Input:** Desired Maximum Iteration *maxIter*

**Output:** An activation pattern $a_m$ ($< a_1, .., a_i, .., a_m >$) at layer *L-1*, with *m* nodes

Set Regulation Factor $\rho$ between [0, 1];

Set *currentActivation = Starting-Point* (a vector of activation $< a_1, \ldots, a_m >$);

Set *previousActivation = currentActivation*;

Set *PropagateActivation=* CCUAP of *currentActivaiton* to layer L (see Section 3.2.5);

Set *ObservedlActivation* (*A*)= Regulate *PropagateActivation* via Algorithm 1;

Calculate error vector (*e*) at layer *L* using Equation (10);

Set *iter* = 0;

**repeat**

  │ Set *iter = iter + 1*;

  │ **foreach** $a_i$ *in previousActivation* **do**

  │  │ Calculate the delta ($\triangle_{a_i, A_j}$) using Equation (11);

  │  │ Calculate the sum of delta for $a_i$, i.e., $\triangle_{a_i}$, using Equation (12);

  │  │ **if** $\triangle_{a_i} > 0$ **then**

  │  │  │ $a_{temp} = a_i + \triangle_{a_i} * (1 - a_i)$;

  │  │  │ **if** $a_{temp} > 1$ **then**

  │  │  │  │ Assign $a_i = 1$;

  │  │  │ **end**

  │  │  │ **else if** $a_{temp} < 0$ **then**

  │  │  │  │ Assign $a_i^{new} = 0$;

  │  │  │ **end**

  │  │  │ **else**

  │  │  │  │ Assign $a_i^{new} = a_{temp}$;

  │  │  │ **end**

  │  │ **end**

  │  │ **else**

  │  │  │ $a_{temp} = a_i + \triangle_{a_i} * (a_i)$;

  │  │  │ **if** $a_{temp} > 1$ **then**

  │  │  │  │ Assign $a_i^{new} = 1$;

  │  │  │ **end**

  │  │  │ **else if** $a_{temp} < 0$ **then**

  │  │  │  │ Assign $a_i^{new} = 0$;

  │  │  │ **end**

  │  │  │ **else**

  │  │  │  │ Assign $a_i^{new} = a_{temp}$;

  │  │  │ **end**

  │  │ **end**

  │ **end**

  │ Set *currentActivation* $= < a_1^{new}, .., a_i^{new}, .., a_m^{new} >$ (new activation vector at layer *L-1*);

  │ Set *previousActivation = currentActivation*;

  │ Set *PropagateActivation=* CCUAP of *currentActivaiton* to layer L (see Section 3.2.5);

  │ Set *ObservedlActivation* (*A*)= Regulate *PropagateActivation* via Algorithm 1;

  │ Calculate error vector (*e*) at layer *L* using Equation (10);

**until** *iter = maxIter*;

**return** *currentActivation*

---

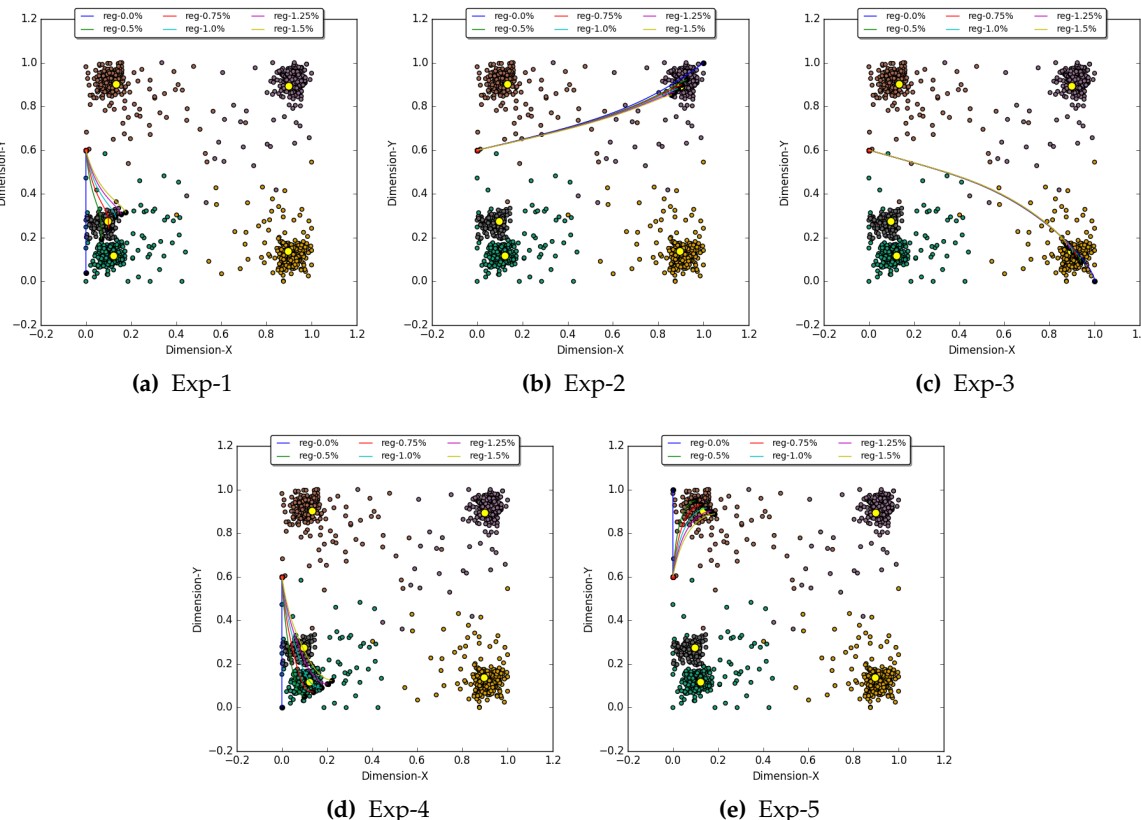

**Figure 8.** The trajectories of activation observed at input layer 0 while carrying out one thousand iterations of the geometric back-propagation (GBP) algorithm. The red circle is the starting point of trajectory, and the black circle is the activation value after the thousandth iteration. The graphs also depict the trajectories observed at input layer 0 with six regulation factors $\rho$ (0%, 0.5%, 0.75%, 1%, 1.25%, and 1.5%). Each graph visualizes the recalled activation for five single-cue recall (SCR) experiments.

**Table 1.** Observations of activations at abstract concept layer 1 for SCR experiments.

| Regulation | Experiment | E-A at Layer 1 | | | | | O-A at Layer 1 | | | | |
|---|---|---|---|---|---|---|---|---|---|---|---|
| % | —— | $C_1$ | $C_2$ | $C_3$ | $C_4$ | $C_5$ | $C_1$ | $C_2$ | $C_3$ | $C_4$ | $C_5$ |
| 0 | Exp-1 | 0 | 0 | 0 | 0 | 1 | 0.08 | 0.61 | 0.07 | 0.01 | 0.47 |
| 0.5 | Exp-1 | 0 | 0 | 0 | 0 | 1 | 0.14 | 0.70 | 0.09 | 0.03 | 0.72 |
| **0.75** | **Exp-1** | 0 | 0 | 0 | 0 | **1** | 0.16 | **0.64** | 0.10 | 0.04 | **0.85** |
| 1 | Exp-1 | 0 | 0 | 0 | 0 | 1 | 0.19 | 0.56 | 0.11 | 0.05 | 0.82 |
| 1.25 | Exp-1 | 0 | 0 | 0 | 0 | 1 | 0.19 | 0.54 | 0.12 | 0.06 | 0.75 |
| 1.5 | Exp-1 | 0 | 0 | 0 | 0 | 1 | 0.20 | 0.52 | 0.12 | 0.06 | 0.71 |
| 0 | Exp-2 | 0 | 0 | 0 | 1 | 0 | 0.08 | 0.01 | 0.08 | 0.61 | 0.02 |
| 0.5 | Exp-2 | 0 | 0 | 0 | 1 | 0 | 0.10 | 0.02 | 0.11 | 0.83 | 0.03 |
| **0.75** | **Exp-2** | 0 | 0 | 0 | **1** | 0 | 0.11 | 0.02 | 0.11 | **0.93** | 0.04 |
| 1 | Exp-2 | 0 | 0 | 0 | 1 | 0 | 0.12 | 0.03 | 0.12 | 0.90 | 0.04 |
| 1.25 | Exp-2 | 0 | 0 | 0 | 1 | 0 | 0.12 | 0.03 | 0.13 | 0.83 | 0.05 |
| 1.5 | Exp-2 | 0 | 0 | 0 | 1 | 0 | 0.13 | 0.03 | 0.13 | 0.78 | 0.05 |
| 0 | Exp-3 | 0 | 0 | 1 | 0 | 0 | 0.01 | 0.07 | 0.57 | 0.07 | 0.06 |
| 0.5 | Exp-3 | 0 | 0 | 1 | 0 | 0 | 0.02 | 0.09 | 0.76 | 0.10 | 0.08 |
| **0.75** | **Exp-3** | 0 | 0 | **1** | 0 | 0 | 0.02 | 0.10 | **0.84** | 0.11 | 0.09 |
| 1 | Exp-3 | 0 | 0 | 1 | 0 | 0 | 0.03 | 0.11 | 0.93 | 0.11 | 0.09 |
| 1.25 | Exp-3 | 0 | 0 | 1 | 0 | 0 | 0.03 | 0.11 | 0.91 | 0.12 | 0.10 |
| 1.5 | Exp-3 | 0 | 0 | 1 | 0 | 0 | 0.03 | 0.12 | 0.84 | 0.13 | 0.10 |
| 0 | Exp-4 | 0 | 1 | 0 | 0 | 0 | 0.06 | 0.57 | 0.07 | 0.01 | 0.42 |
| 0.5 | Exp-4 | 0 | 1 | 0 | 0 | 0 | 0.08 | 0.75 | 0.10 | 0.02 | 0.50 |
| **0.75** | **Exp-4** | 0 | **1** | 0 | 0 | 0 | 0.09 | **0.80** | 0.12 | 0.02 | **0.52** |
| 1 | Exp-4 | 0 | 1 | 0 | 0 | 0 | 0.10 | 0.77 | 0.13 | 0.03 | 0.53 |
| 1.25 | Exp-4 | 0 | 1 | 0 | 0 | 0 | 0.10 | 0.73 | 0.14 | 0.03 | 0.54 |
| 1.5 | Exp-4 | 0 | 1 | 0 | 0 | 0 | 0.11 | 0.69 | 0.15 | 0.04 | 0.54 |

**Table 1.** *Cont.*

| Regulation | Experiment | E-A at Layer 1 | | | | | O-A at Layer 1 | | | | |
|---|---|---|---|---|---|---|---|---|---|---|---|
| % | ——— | $C_1$ | $C_2$ | $C_3$ | $C_4$ | $C_5$ | $C_1$ | $C_2$ | $C_3$ | $C_4$ | $C_5$ |
| 0 | Exp-5 | 1 | 0 | 0 | 0 | 0 | 0.58 | 0.07 | 0.01 | 0.07 | 0.13 |
| 0.5 | Exp-5 | 1 | 0 | 0 | 0 | 0 | 0.78 | 0.09 | 0.02 | 0.10 | 0.15 |
| 0.75 | Exp-5 | 1 | 0 | 0 | 0 | 0 | 0.85 | 0.10 | 0.02 | 0.11 | 0.16 |
| **1** | **Exp-5** | **1** | **0** | **0** | **0** | **0** | **0.88** | **0.10** | **0.03** | **0.12** | **0.17** |
| 1.25 | Exp-5 | 1 | 0 | 0 | 0 | 0 | 0.84 | 0.11 | 0.03 | 0.13 | 0.17 |
| 1.5 | Exp-5 | 1 | 0 | 0 | 0 | 0 | 0.79 | 0.11 | 0.04 | 0.13 | 0.18 |

E-A, expected activation, O-A, observed activation.

### 3.5.2. Multiple-Cue Recall (MCR) Experiment

The MCR experiments were carried out to determine the recall vector at input layer 0 for an E-A vector at layer 1. The constituents of the E-A vector are degree of confidence (DoC) values that define the expected representation of each abstract concept node at layer 1. To demonstrate MCR, five experiments were performed, and in every experiment, six regulation factors $\rho$ (0%, 0.1%, 0.2%, 0.3%, 0.4%, and 0.5%) were used to make inferences. Table 2 lists the observations of O-A for the respective E-A in each MCR experiment. Figure 9 displays the trajectories of recalled activation in layer 0 for six regulation factors with respect to each experiment. The E-A vectors used in the MCR experiments are vectors obtained by propagating an input activation from layer 0 to layer 1. Hence, in MCR simulation, we also have an expected recall (E-R) vector to perform evaluations. The following are the MCR experiment descriptions along with observations:

- **Exp-6:** In this experiment, an E-A vector of *[0.57, 0.16, 0.06, 0.15, 0.25]* was provided to the GBP algorithm. With this E-A, we wanted to recall activation at input layer 0, that is 57%, 16%, 06%, 15% and 25% represented by nodes $C_1$, $C_2$, $C_3$, $C_4$, and $C_5$, respectively. Table 2 lists the O-A observed for the six regulation factors. The results with $\rho$ of 0% and 0.1% show the outcome, which is almost identical to that of E-A. The E-R vector of this experiment was *[0.2256, 0.7610]*. With $\rho$ of 0% and 0.1%, the observed recall was *[0.2260781, 0.7647118]* and *[0.2343844, 0.7602842]*, respectively, which are also similar to the expected recall vector. Figure 9a shows the trajectories of all the recalled activation vectors at layer 0 for the E-A vector w.r.t. their six regulation factors.

- **Exp-7:** For this experiment, the GBP algorithm was injected with an E-A vector of *[0.07, 0.04, 0.23, 0.46, 0.05]* for recalling an activation vector at input layer 0. The six O-A vectors obtained for the six regulation factors can be seen in Table 2. The O-A vector for $\rho$ of 0% and 0.1% was almost the same as the E-A vector. The recalled activations regulation factor 0% was *[0.9875402, 0.6551013]*, which is almost similar to the E-R vector *[0.9896, 0.6568]*. Figure 9b shows all the recalled trajectories for this experiment.

- **Exp-8:** In this experiment, the GBP algorithm was initialized with the E-A vector of *[0.09, 0.5, 0.22, 0.05, 0.52]*. The E-R vector for this experiment was *[0.3458, 0.1157]*, and two similar vectors, *[0.3444873, 0.1032499]* and *[0.3489568, 0.1284956]*, were recalled in this experiment using regulation of 0% and 0.1%, respectively. Figure 9c shows all recalled trajectories. The O-A vectors obtained with the regulation of 0% and 0.1% were also identical to the E-A vector of this experiment—see Table 2.

- **Exp-9:** The recall simulation in this experiment was instantiated with an E-A vector of *[0.09, 0.40, 0,28, 0.07, 0.35]*, and a recall vector of *[0.4410, 0.1341]* was expected at input layer 0. Upon using the GBP algorithm with six regulation factors, the recall operation without regulation, i.e., $\rho$ of 0%, produced the most similar recall vector *[0.4370989, 0.1308456]* and the corresponding O-A vector at layer 1. However, the outcome with 0.1% regulations was also similar to a recalled vector *[0.4392012, 0.1518065]* at input layer 0—see Figure 9d.

- **Exp-10:** This experiment used an E-A vector *[0.07, 0.09, 0.44, 0.26, 0.10]* in order to obtain an E-R vector *[0.8813, 0.4145]* at input layer 0. The six simulations were carried

out with different regulation factors, and it was observed that the results with $\rho$ of 0% and 0.1% produced results very near those of the E-R, i.e., *[0.8873921, 0.4137484]* and *[0.8702277, 0.4153301]*, respectively—see Figure 9e for all trajectories. The same observations were made at the O-A vectors for $\rho$ of 0% and 0.1% at layer 1—see Table 2.

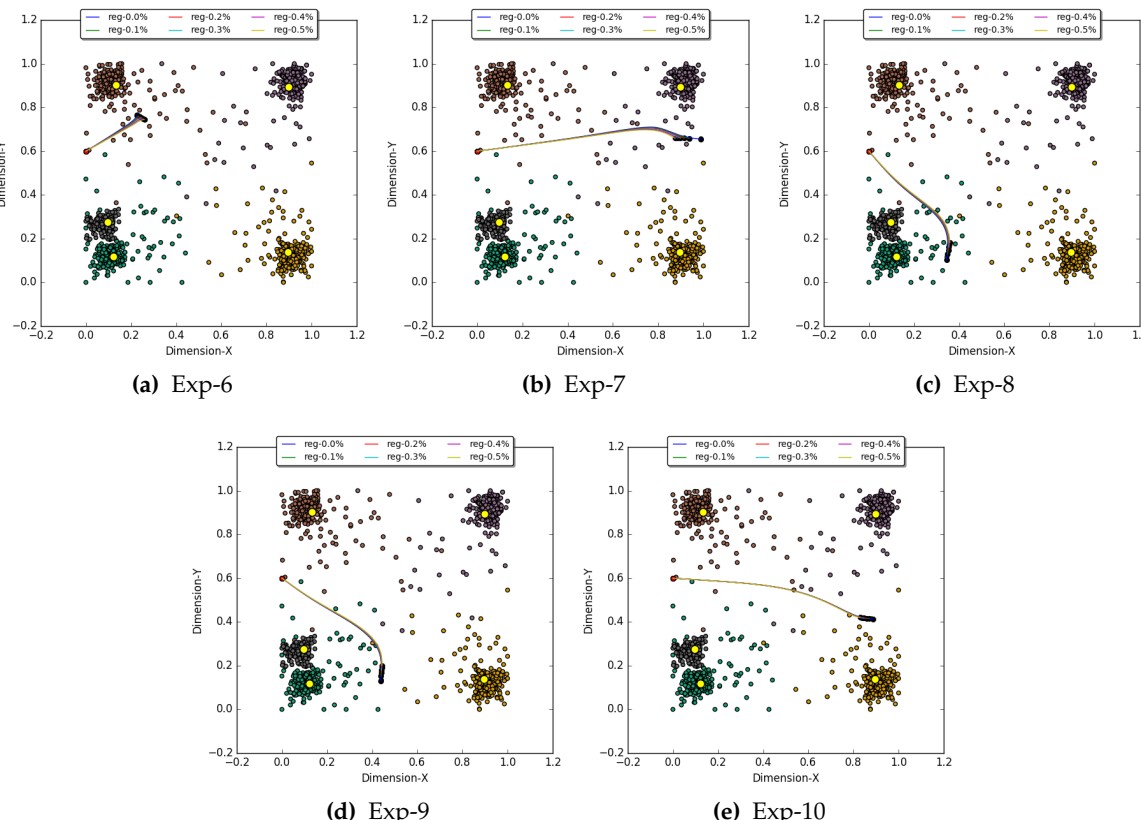

**(a)** Exp-6      **(b)** Exp-7      **(c)** Exp-8

**(d)** Exp-9      **(e)** Exp-10

**Figure 9.** The trajectories of activation observed at input layer 0 while carrying out one thousand iterations of the GBP algorithm. The red circle is the starting point of trajectory, and the black circle is the activation value after the thousandth iteration. The graphs also depict the trajectories observed at input layer 0 with six regulation factors $\rho$ (0%, 0.1%, 0.2%, 0.3%, 0.4%, and 0.5%). Each graph visualizes the recalled activation for five multiple-cue recall (MCR) experiments.

**Table 2.** Observations of activations at abstract concept layer 1 for MCR experiments.

| Regulation | Experiment | E-A at Layer 1 | | | | | O-A at Layer 1 | | | | |
|---|---|---|---|---|---|---|---|---|---|---|---|
| % | ——— | $C_1$ | $C_2$ | $C_3$ | $C_4$ | $C_5$ | $C_1$ | $C_2$ | $C_3$ | $C_4$ | $C_5$ |
| **0** | **Exp-6** | 0.57 | 0.16 | 0.06 | 0.15 | 0.25 | 0.579 | 0.162 | 0.063 | 0.148 | 0.246 |
| **0.1** | **Exp-6** | 0.57 | 0.16 | 0.06 | 0.15 | 0.25 | 0.567 | 0.163 | 0.066 | 0.151 | 0.247 |
| 0.2 | Exp-6 | 0.57 | 0.16 | 0.06 | 0.15 | 0.25 | 0.556 | 0.164 | 0.068 | 0.154 | 0.249 |
| 0.3 | Exp-6 | 0.57 | 0.16 | 0.06 | 0.15 | 0.25 | 0.546 | 0.166 | 0.070 | 0.156 | 0.250 |
| 0.4 | Exp-6 | 0.57 | 0.16 | 0.06 | 0.15 | 0.25 | 0.537 | 0.167 | 0.072 | 0.159 | 0.251 |
| 0.5 | Exp-6 | 0.57 | 0.16 | 0.06 | 0.15 | 0.25 | 0.529 | 0.168 | 0.074 | 0.161 | 0.252 |
| **0** | **Exp-7** | 0.07 | 0.04 | 0.23 | 0.46 | 0.05 | 0.072 | 0.039 | 0.233 | 0.465 | 0.051 |
| **0.1** | **Exp-7** | 0.07 | 0.04 | 0.23 | 0.46 | 0.05 | 0.086 | 0.048 | 0.235 | 0.482 | 0.061 |
| 0.2 | Exp-7 | 0.07 | 0.04 | 0.23 | 0.46 | 0.05 | 0.093 | 0.052 | 0.235 | 0.486 | 0.067 |
| 0.3 | Exp-7 | 0.07 | 0.04 | 0.23 | 0.46 | 0.05 | 0.099 | 0.055 | 0.236 | 0.487 | 0.071 |
| 0.4 | Exp-7 | 0.07 | 0.04 | 0.23 | 0.46 | 0.05 | 0.103 | 0.058 | 0.236 | 0.487 | 0.074 |
| 0.5 | Exp-7 | 0.07 | 0.04 | 0.23 | 0.46 | 0.05 | 0.107 | 0.061 | 0.236 | 0.485 | 0.077 |
| **0** | **Exp-8** | 0.09 | 0.50 | 0.22 | 0.05 | 0.42 | 0.091 | 0.502 | 0.218 | 0.051 | 0.414 |
| **0.1** | **Exp-8** | 0.09 | 0.50 | 0.22 | 0.05 | 0.42 | 0.099 | 0.497 | 0.221 | 0.056 | 0.424 |
| 0.2 | Exp-8 | 0.09 | 0.50 | 0.22 | 0.05 | 0.42 | 0.105 | 0.492 | 0.223 | 0.061 | 0.430 |
| 0.3 | Exp-8 | 0.09 | 0.50 | 0.22 | 0.05 | 0.42 | 0.110 | 0.486 | 0.224 | 0.064 | 0.434 |
| 0.4 | Exp-8 | 0.09 | 0.50 | 0.22 | 0.05 | 0.42 | 0.114 | 0.481 | 0.225 | 0.067 | 0.437 |
| 0.5 | Exp-8 | 0.09 | 0.50 | 0.22 | 0.05 | 0.42 | 0.117 | 0.476 | 0.226 | 0.070 | 0.439 |
| **0** | **Exp-9** | 0.09 | 0.40 | 0.28 | 0.07 | 0.35 | 0.090 | 0.401 | 0.280 | 0.070 | 0.350 |
| **0.1** | **Exp-9** | 0.09 | 0.40 | 0.28 | 0.07 | 0.35 | 0.096 | 0.397 | 0.281 | 0.076 | 0.355 |
| 0.2 | Exp-9 | 0.09 | 0.40 | 0.28 | 0.07 | 0.35 | 0.101 | 0.394 | 0.282 | 0.080 | 0.358 |
| 0.3 | Exp-9 | 0.09 | 0.40 | 0.28 | 0.07 | 0.35 | 0.105 | 0.391 | 0.282 | 0.084 | 0.360 |

**Table 2.** *Cont.*

| Regulation | Experiment | E-A at Layer 1 | | | | | O-A at Layer 1 | | | | |
|---|---|---|---|---|---|---|---|---|---|---|---|
| % | ——— | $C_1$ | $C_2$ | $C_3$ | $C_4$ | $C_5$ | $C_1$ | $C_2$ | $C_3$ | $C_4$ | $C_5$ |
| 0.4 | Exp-9 | 0.09 | 0.40 | 0.28 | 0.07 | 0.35 | 0.109 | 0.388 | 0.282 | 0.087 | 0.362 |
| 0.5 | Exp-9 | 0.09 | 0.40 | 0.28 | 0.07 | 0.35 | 0.112 | 0.385 | 0.281 | 0.090 | 0.363 |
| **0** | **Exp-10** | 0.07 | 0.09 | 0.44 | 0.26 | 0.10 | 0.068 | 0.093 | 0.440 | 0.263 | 0.099 |
| **0.1** | **Exp-10** | 0.07 | 0.09 | 0.44 | 0.26 | 0.10 | 0.073 | 0.098 | 0.437 | 0.264 | 0.105 |
| 0.2 | Exp-10 | 0.07 | 0.09 | 0.44 | 0.26 | 0.10 | 0.076 | 0.103 | 0.434 | 0.264 | 0.110 |
| 0.3 | Exp-10 | 0.07 | 0.09 | 0.44 | 0.26 | 0.10 | 0.079 | 0.106 | 0.431 | 0.264 | 0.114 |
| 0.4 | Exp-10 | 0.07 | 0.09 | 0.44 | 0.26 | 0.10 | 0.082 | 0.109 | 0.428 | 0.265 | 0.117 |
| 0.5 | Exp-10 | 0.07 | 0.09 | 0.44 | 0.26 | 0.10 | 0.084 | 0.112 | 0.425 | 0.265 | 0.120 |

E-A, expected activation, O-A, observed activation.

### 3.5.3. Discussion

The experiments in Sections 3.5.1 and 3.5.2 demonstrate a notable behavior of RAN by simulating the cued recall operation through a Toy-data problem. The intra-Layer learning (i.e., CSRL) is uniquely utilized by RAN modeling to interpret the association among the concepts as inhibitory, excitatory, or neutral. Furthermore, the intra-layer regulation (Algorithm 1) uses intra-layer learning (CSRL) and its interpretations to produce a regulatory effect over the activation of the concepts (at the same layer). The geometric back-propagation operation (Algorithm 2) is a method analogous to remembering something learned in an abstract form and recalling its concrete features. For example, while remembering the abstract concept "house", we recall concrete features related to the house, such as "mother", "father", "wife", and "pets".

In the graphs in Figures 8 and 9, we can see that all the trajectories commence from a starting point (red dot) and converge to a point after one thousand iterations. Each point in a trajectory represents a temporal mental state while recalling a concrete concept. Every time a concrete concept (activation vector in layer 0) is recalled, its corresponding abstract concept (at layer 1) is compared with the expected abstract concepts. The difference between expected and observed activation is propagated back as the error to the previously recalled activations at layer 0. In the next instance, the corrected recalled activation at layer 0 repeats the process until one thousand iterations are completed.

It was observed that without regulation, i.e., 0% $\rho$, the trajectory converges to a point but with a minimal amount of regulation, and the result improves. For instance, in the graphs in Figure 8, only one abstract concept was being recalled and the results improved when the regulation was introduced. In the two experiments (SCR and MCR), we can see that the two different sets of regulation factors are considered. These sets were obtained empirically, but we can see that the set of the regulation factors for the SCR experiment has a higher value. This is because the GBP algorithm strives to minimizes the error at each abstract concept node at layer 1, and in the geometrical context, similarity cannot be the same for more than one abstract concept. Thus, the trajectory converges to a point, but the result improves when a minimal amount of regulation is induced. In the MCR experiment, the best outcome is observed with little or no regulation because the expected similarity (DoC and E-A) is a non-zero value. The other reason is that these are possible expected similarity vectors, unlike those in the SCR experiments.

## 4. Cued Recall Demonstration with MNIST Data

The MNIST [70] dataset is a collection of handwritten images of digits (0, 1, 2, 3, 4, 5, 6, 7, 8 and 9), where each image is black and white in color and has a $28 \times 28$ pixel size. This dataset of image domains is used to demonstrate the Ccued recall operation of learned abstract concepts representing different digits. Two types of investigations were conducted with this dataset: first, multiple binary valued cue recall (MBVCR), where the E-A vector is a binary value ([0, 1]) vector, and second, multiple-cue recall (MCR).

For this experiment, one thousand images were selected randomly from the MNIST dataset. The $28 \times 28$ image was transformed in a single vector of 784 attributes, where each attribute corresponds to a pixel of the image. Additionally, the attribute values of the

data were normalized between 0 and 1 using min–max normalization (black pixel is min, i.e., 0, and white pixel is max, i.e., 255). Having preprocessed the data, the RAN modeling procedure was instantiated by selecting the K-mean clustering algorithm as the concept identifier. *K* was initialized with 30 to determine thirty categories in the input space. The model was configured to grow one level deep and build convex abstract concept (CAC) layer 1. After carrying out all four steps of RAN modeling (see Section 3.2), a model was obtained—see Figure 10.

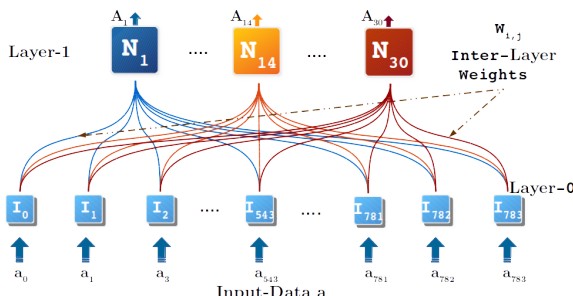

**Figure 10.** RAN model generated with MNIST dataset.

In Figure 10, layer 0 has 784 nodes representing each pixel; CAC layer 1 has 30 nodes representing the thirty categories identified during the CCI process in RAN modeling. The inter-layer weights (ILWs) are the cluster centers (CRDPs) of the thirty clusters. Figure 11 shows the ILWs reconstructed in image form of $28 \times 28$ pixels. In RAN modeling, a CRDP is the optimum representative of an input level category at CAC layer 1. Therefore, Figure 11a–ad are the best represented by CAC node $N_1$, ..., $N_{30}$, respectively. In Figure 11 it is noticeable that each digit is represented by at least two CAC node of layer 1. The digit 9 is represented by the largest number of nodes, i.e., $N_2$, $N_{15}$, $N_{18}$, and $N_{24}$. In contrast, digit 4 is represented by two nodes, $N_1$ and $N_{10}$. Figure 11d,m,u,x show that the CAC nodes $N_4$, $N_{13}$, $N_{21}$, and $N_{24}$ do not represent an individual digit. Node $N_4$ and $N_{13}$ jointly represent digits 3 and 8; node $N_{21}$ looks like two digits, 3 and 5; and $N_{24}$ depicts digits 7 and 9.

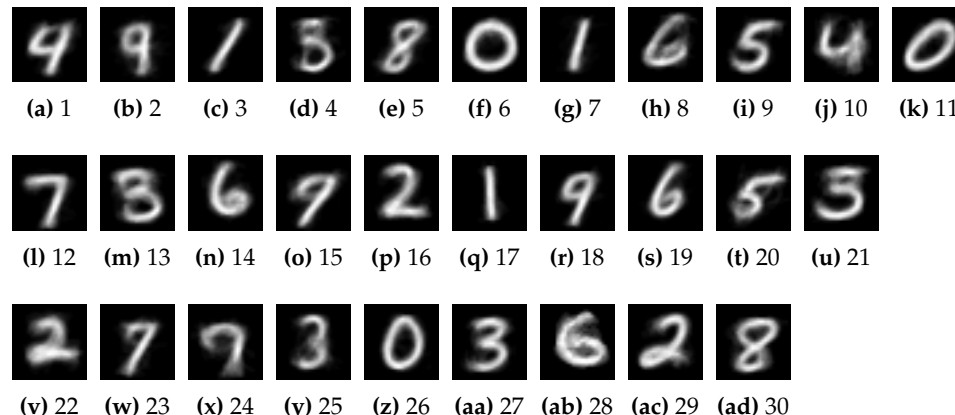

**(a)** 1 **(b)** 2 **(c)** 3 **(d)** 4 **(e)** 5 **(f)** 6 **(g)** 7 **(h)** 8 **(i)** 9 **(j)** 10 **(k)** 11

**(l)** 12 **(m)** 13 **(n)** 14 **(o)** 15 **(p)** 16 **(q)** 17 **(r)** 18 **(s)** 19 **(t)** 20 **(u)** 21

**(v)** 22 **(w)** 23 **(x)** 24 **(y)** 25 **(z)** 26 **(aa)** 27 **(ab)** 28 **(ac)** 29 **(ad)** 30

**Figure 11.** The thirty CRDPs (cluster centers). Each node in layer 1 of Figure 10 acts as the abstract representative of each CRDP.

For simplicity, Figure 10 shows only inter-layer learning; intra-Layer learning (CSRL weights) was also performed on both the input layer 0 and CAC layer 1. The CSRL weights at input layer 0 were a $784 \times 784$ matrix, and at CAC layer 1, a $30 \times 30$ matrix was learned. These two intra-layer learning procedures were utilized by the GBP algorithm to simulate the recall operations. In all of the experiments, the GBP algorithm was configured to iterate five hundred times. The GBP algorithm was initialized with a vector with activation 1 for all 784 nodes of input layer 0. The image at Iter-0 (see Tables 3 and 4) is white because activation 1 corresponds to pixel value of 255 depicting white color.

**Table 3.** Intuitive multiple binary valued cue recall (MBVCR) observations with RAN model of MNIST data.

| Digit | æ | Iter ⇒ | 0 | 3 | 5 | 8 | 11 | 19 | 25 | 35 | 41 | 71 | 81 | 91 | 101 | 151 | 201 | 251 | 301 | 351 | 401 | 451 | 501 |
|---|---|---|---|---|---|---|---|---|---|---|---|---|---|---|---|---|---|---|---|---|---|---|---|
| 0 | 0% | | | | | | | | | | | | | | | | | | | | | | |
| 0 | 0.009% | | | | | | | | | | | | | | | | | | | | | | |
| 1 | 0% | | | | | | | | | | | | | | | | | | | | | | |
| 1 | 0.009% | | | | | | | | | | | | | | | | | | | | | | |
| 2 | 0% | | | | | | | | | | | | | | | | | | | | | | |
| 2 | 0.009% | | | | | | | | | | | | | | | | | | | | | | |
| 3 | 0% | | | | | | | | | | | | | | | | | | | | | | |
| 3 | 0.009% | | | | | | | | | | | | | | | | | | | | | | |
| 4 | 0% | | | | | | | | | | | | | | | | | | | | | | |
| 4 | 0.009% | | | | | | | | | | | | | | | | | | | | | | |
| 5 | 0% | | | | | | | | | | | | | | | | | | | | | | |
| 5 | 0.009% | | | | | | | | | | | | | | | | | | | | | | |
| 6 | 0% | | | | | | | | | | | | | | | | | | | | | | |
| 6 | 0.009% | | | | | | | | | | | | | | | | | | | | | | |
| 7 | 0% | | | | | | | | | | | | | | | | | | | | | | |
| 7 | 0.009% | | | | | | | | | | | | | | | | | | | | | | |
| 8 | 0% | | | | | | | | | | | | | | | | | | | | | | |
| 8 | 0.009% | | | | | | | | | | | | | | | | | | | | | | |
| 9 | 0% | | | | | | | | | | | | | | | | | | | | | | |
| 9 | 0.009% | | | | | | | | | | | | | | | | | | | | | | |

**Table 4.** Observations of multiple-cue recall operation with RAN model of MNIST data.

| Digit | æ | Iter ⇒ | 0 | 3 | 5 | 8 | 11 | 19 | 25 | 35 | 41 | 71 | 81 | 91 | 101 | 151 | 201 | 251 | 301 | 351 | 401 | 451 | 501 |
|---|---|---|---|---|---|---|---|---|---|---|---|---|---|---|---|---|---|---|---|---|---|---|---|
| 0 | 0% | | | | | | | | | | | | | | | | | | | | | | |
| 0 | 0.009% | | | | | | | | | | | | | | | | | | | | | | |
| 1 | 0% | | | | | | | | | | | | | | | | | | | | | | |
| 1 | 0.009% | | | | | | | | | | | | | | | | | | | | | | |
| 2 | 0% | | | | | | | | | | | | | | | | | | | | | | |
| 2 | 0.009% | | | | | | | | | | | | | | | | | | | | | | |
| 3 | 0% | | | | | | | | | | | | | | | | | | | | | | |
| 3 | 0.009% | | | | | | | | | | | | | | | | | | | | | | |
| 4 | 0% | | | | | | | | | | | | | | | | | | | | | | |
| 4 | 0.009% | | | | | | | | | | | | | | | | | | | | | | |
| 5 | 0% | | | | | | | | | | | | | | | | | | | | | | |
| 5 | 0.009% | | | | | | | | | | | | | | | | | | | | | | |
| 6 | 0% | | | | | | | | | | | | | | | | | | | | | | |
| 6 | 0.009% | | | | | | | | | | | | | | | | | | | | | | |
| 7 | 0% | | | | | | | | | | | | | | | | | | | | | | |
| 7 | 0.009% | | | | | | | | | | | | | | | | | | | | | | |
| 8 | 0% | | | | | | | | | | | | | | | | | | | | | | |
| 8 | 0.009% | | | | | | | | | | | | | | | | | | | | | | |
| 9 | 0% | | | | | | | | | | | | | | | | | | | | | | |
| 9 | 0.009% | | | | | | | | | | | | | | | | | | | | | | |

In each experiment, the two cued recall demonstrations, MBVCR and MCR, use the expected activating (E-A) vector as listed in Table 5. The experiments of single digits and

combined digits for the MBVCR operation used an E-A vector of binary values, where binary 1 at a node N was assigned w.r.t the digit(s) being recalled. For instance, the E-A vector of digit 2 was formed by initializing the E-A vector with binary 1 for nodes $N_{16}$, $N_{22}$, and $N_{29}$ and binary 0 for the remaining 27 CAC nodes (see Table 5). The E-A vectors of MCR experiments are the actual activation values obtained by propagating the inter-layer weights upwards (CRDPs, see Figure 11) as input. The weights represented by Figure 11a–c,e,f,i,l–n,p were provided as input to the CCUAP operation to observe their respective activation at CAC layer 1. These observed activation vectors were used as E-A for each digit recall operation (see last ten MCR E-As in Table 5).

**Table 5.** Expected activation injected at thirty convex abstract concept (CAC) nodes in layer 1 of RAN model for MNIST data.

| Digit | Exp | Expected Activation (E-A) |
|---|---|---|
| 0 | MBVCR | [ 0,0,0,0,0,1,0,0,0,0,1,0,0,0,0,0,0,0,0,0,0,0,0,0,0,1,0,0,0,0 ] |
| 1 | MBVCR | [ 0,0,1,0,0,0,1,0,0,0,0,0,0,0,0,0,0,1,0,0,0,0,0,0,0,0,0,0,0,0 ] |
| 2 | MBVCR | [ 0,0,0,0,0,0,0,0,0,0,0,0,0,0,0,0,1,0,0,0,0,0,1,0,0,0,0,0,0,1,0 ] |
| 3 | MBVCR | [ 0,0,0,1,0,0,0,0,0,0,0,0,1,0,0,0,0,0,0,0,0,0,0,0,0,1,0,1,0,0,0 ] |
| 4 | MBVCR | [ 1,0,0,0,0,0,0,0,1,0,0,0,0,0,0,0,0,0,0,0,0,0,0,0,0,0,0,0,0,0 ] |
| 5 | MBVCR | [ 0,0,0,0,0,0,0,0,1,0,0,0,0,0,0,0,0,0,0,1,1,0,0,0,0,0,0,0,0,0 ] |
| 6 | MBVCR | [ 0,0,0,0,0,0,1,0,0,0,0,0,1,0,0,0,0,1,0,0,0,0,0,0,0,0,0,1,0,0 ] |
| 7 | MBVCR | [ 0,0,0,0,0,0,0,0,0,1,0,0,0,0,0,0,0,0,0,0,0,0,1,0,0,0,0,0,0,0 ] |
| 8 | MBVCR | [ 0,0,0,0,1,0,0,0,0,0,0,0,0,0,0,0,0,0,0,0,0,0,0,0,0,0,0,0,0,1 ] |
| 9 | MBVCR | [ 0,1,0,0,0,0,0,0,0,0,0,0,0,1,0,0,1,0,0,0,0,0,1,0,0,0,0,0,0,0 ] |
| 2 and 5 | MBVCR | [ 0,0,0,0,0,0,0,0,1,0,0,0,0,0,0,0,1,0,0,0,1,1,1,0,0,0,0,0,0,1,0 ] |
| 3 and 5 | MBVCR | [ 0,0,0,1,0,0,0,0,1,0,0,0,1,0,0,0,0,0,0,1,1,0,0,0,0,1,0,1,0,0,0 ] |
| 0 and 1 | MBVCR | [ 0,0,1,0,0,1,1,0,0,0,1,0,0,0,0,1,0,0,0,0,0,0,0,0,1,0,0,0,0,0 ] |
| 0 | MCR | [ 0.26,0.25,0.22,0.30,0.27,1.00,0.23,0.30,0.28,0.30,0.35,0.26,0.32,0.26,0.25,0.28,0.22,0.23,0.23,0.25,0.33,0.28,0.25,0.29,0.28,0.39,0.28,0.31,0.27,0.30 ] |
| 1 | MCR | [ 0.29,0.37,1.00,0.35,0.43,0.22,0.49,0.33,0.34,0.32,0.26,0.31,0.31,0.28,0.34,0.39,0.38,0.35,0.34,0.39,0.31,0.35,0.43,0.31,0.42,0.27,0.36,0.24,0.31,0.33 ] |
| 2 | MCR | [ 0.28,0.35,0.39,0.38,0.40,0.28,0.38,0.36,0.33,0.31,0.26,0.29,0.36,0.35,0.29,1.00,0.34,0.31,0.37,0.34,0.36,0.43,0.34,0.29,0.47,0.32,0.37,0.33,0.37,0.34 ] |
| 3 | MCR | [ 0.31,0.38,0.31,0.46,0.35,0.32,0.32,0.31,0.35,0.33,0.26,0.32,1.00,0.33,0.31,0.36,0.31,0.33,0.31,0.30,0.40,0.34,0.33,0.35,0.38,0.34,0.46,0.35,0.27,0.37 ] |
| 4 | MCR | [ 1.00,0.46,0.29,0.32,0.38,0.26,0.30,0.38,0.37,0.44,0.31,0.42,0.31,0.39,0.44,0.28,0.31,0.48,0.36,0.37,0.29,0.34,0.37,0.42,0.34,0.34,0.34,0.31,0.32,0.40 ] |
| 5 | MCR | [ 0.37,0.42,0.34,0.44,0.47,0.28,0.36,0.41,1.00,0.29,0.39,0.36,0.35,0.34,0.37,0.33,0.36,0.42,0.43,0.40,0.42,0.33,0.38,0.35,0.43,0.37,0.45,0.29,0.33,0.41 ] |
| 6 | MCR | [ 0.39,0.40,0.28,0.38,0.33,0.26,0.31,0.45,0.34,0.38,0.28,0.31,0.33,1.00,0.34,0.35,0.32,0.36,0.42,0.32,0.32,0.40,0.33,0.35,0.35,0.36,0.34,0.42,0.33,0.33 ] |
| 7 | MCR | [ 0.42,0.45,0.31,0.34,0.35,0.26,0.34,0.32,0.36,0.34,0.28,1.00,0.32,0.31,0.41,0.29,0.35,0.47,0.33,0.35,0.29,0.32,0.43,0.47,0.38,0.30,0.34,0.25,0.28,0.34 ] |
| 8 | MCR | [ 0.38,0.41,0.43,0.39,1.00,0.27,0.40,0.38,0.47,0.32,0.33,0.35,0.35,0.33,0.40,0.40,0.37,0.44,0.42,0.44,0.38,0.39,0.45,0.32,0.44,0.34,0.44,0.29,0.39,0.48 ] |
| 9 | MCR | [ 0.46,1.00,0.37,0.40,0.41,0.25,0.39,0.38,0.42,0.38,0.29,0.45,0.38,0.40,0.40,0.35,0.39,0.51,0.40,0.39,0.32,0.40,0.43,0.44,0.43,0.32,0.44,0.32,0.30,0.40 ] |

## 4.1. Multiple Binary Valued Cue Recall (MBVCR) Operation

For the MBVCR operation, the RAN model generated with MNIST data (see Figure 10) was used in order to obtain the recalled activation at input layer 0 for a given expected activation vector at CAC layer 1. As described earlier, the E-A vector for MBVCR is a vector of binary values, which is provided as input to the GBP algorithm to perform the recall operation. The experiments themselves are divided into two categories, i.e., intuitive and non-intuitive recall.

### 4.1.1. Intuitive MBVCR Experiment

In this experiment, by intuition, we hypothesize that if all CAC nodes (representing a digit) are activated with value 1, then its recall at layer 0 must depict that digit. For example, if the CAC nodes $N_6$, $N_{11}$, and $N_{26}$ (see Figure 11) are activated with a value of 1 (and 0 for others), then we should obtain an image depicting a blend of zero digits after the recall operation. We performed this intuitive recall experiment for all ten digits. The binary E-A vector of all ten digits for the intuitive MBVCR operation is listed in Table 5. Table 3 displays the recalled images of all twenty experiments. For every digit, two investigations were made; the first without regulation, i.e., $\rho = 0$; the second with a regulation of 0.009%.

The first observation is that there is a very insignificant difference between the images recalled with and without regulation. After the second iteration, the digit being recalled begins to appear. Beyond the 80th iteration, no significant change is observed in the recalled images. The recalled images of digits 0, 1, 2, 3, 7, and 8 are recognizable after the 500th iteration. However, the digits 4, 5, and 9 are not very discernible in their last iteration; this is because these digits are cross-represented by CAC nodes (see Figure 11). All the images recalled in this experiment contain noise (i.e., the gray shades), because the E-A vector has two values, either 0 or 1, and a node can be 100% similar to only one other node. Therefore, the GBP algorithm adjusts the activation at the CAC node such that the best representation of the E-A is achieved.

### 4.1.2. Non-Intuitive MBVCR Experiment

In these experiments, the E-A vector contains an activation value of 1 for CAC nodes representing two different digits. The objective of the experiment was to determine what is recalled at the input layer 0 when the CAC nodes, representing two different digits, expect high activation. The three E-As used in this experiment are a combination of activation $2^s$-with-$5^s$, $3^s$-with-$5^s$, and $0^s$-with-$1^s$ (see Table 5 for E-A vectors with the coupled digits). The observations without regulation and with regulation are similar—see Table 6. The blend of the $2^s$-with-$5^s$ recalls an image that looks like the letter x. The fusion of $3^s$-with-$5^s$ recalls an image similar to the digit 3. The combination of $0^s$-with-$1^s$ in the beginning looked like the symbol $\Phi$, but this was distorted later. It is also observed that the images obtained after all the iterations had less noise when compared to the those of the intuitive MBVCR experiments. This is probably because a number of CAC nodes were expecting activation, i.e., more cues were provided.

**Table 6.** Non-intuitive MBVCR observations with RAN model of MNIST data.

| Digit | æ | Iter ⇒ | 0 | 3 | 5 | 8 | 11 | 19 | 25 | 35 | 41 | 71 | 81 | 91 | 101 | 151 | 201 | 251 | 301 | 351 | 401 | 451 | 501 |
|---|---|---|---|---|---|---|---|---|---|---|---|---|---|---|---|---|---|---|---|---|---|---|---|
| 2 and 5 | 0% | | | | | | | | | | | | | | | | | | | | | | |
| 2 and 5 | 0.009% | | | | | | | | | | | | | | | | | | | | | | |
| 3 and 5 | 0% | | | | | | | | | | | | | | | | | | | | | | |
| 3 and 5 | 0.009% | | | | | | | | | | | | | | | | | | | | | | |
| 0 and 1 | 0% | | | | | | | | | | | | | | | | | | | | | | |
| 0 and 1 | 0.009% | | | | | | | | | | | | | | | | | | | | | | |

### 4.2. Multiple-Cue Recall (MCR) Experiment

This experiment is the same as the experiment discussed in Section 3.5.2. The E-A vectors are the activation values observed at CAC nodes by propagating the inter-layer weights using the CCUAP operation of RAN modeling.

Figure 11 shows the images re-constructed for each inter-layer weight. The E-As corresponding to Figure 11a–c,e,f,i,l–n,p are listed in Table 5 and are used in MCR demonstrations of this section.

The objective of this experiment was the same as that of MBVCR experiments, i.e., obtaining an activation vector at input layer 0 that corresponds to an E-A vector. However, in this experiment, an expected recall (E-R) was already known. Therefore, the E-As of ten digits (see MCR E-As in Table 5) were expected to recall the images in Figure 11a–c,e,f,i,l–n,p.

In this experiment, the observations with and without regulation are identical. It is also worth noting that after the 500th iteration, the recalled images of all ten digits were similar to the E-R images of each digits.

### 4.3. Discussion

There are a few things worth mentioning in the recall demonstrations of RAN modeling with the MNIST dataset. First, we can reconstruct cognizable images of a digit by activating the CAC nodes representing that digit. Second, it is possible to recall both an intuitive and non-intuitive blend of learned abstract concepts (in these experiments, the abstract concepts are a generic representations of digits). Third, the recalled activations, with and without regulation, are similar for a complex dataset like MNIST. Last, the more cues we provide in the E-A vector, the more accurate the recall operation becomes. The recall capability of RAN modeling was applied to the reconsctruction of an encoded image, where the image was encoded using RAN convex concept modeling and reconstruction was performend via the geometric back-propagation (or recall) operation [71].

### 5. Conclusions

Recall is a cognitive process that can also be seen as an act of remembering a concept. Concepts are normally perceived in a hierarchical form, where the concrete concepts occupy

the lower level, and the abstract concepts take up the relatively higher level in the hierarchy. According to context availability theory, the context among the concrete concepts is easily determined when compared to abstract concepts; hence, their comprehension and recall are also difficult. However, if we can relate abstract concepts to one another, it is possible to deduce a contextual relationship among them. In this article, we exploited the intra-layer associations learned among the concepts (including abstract concepts) using RAN modeling to establish context among the concepts. We use this context-related information to induce a regulatory effect on the concepts and further to simulate the recall operations.

To demonstrate the effect of regulation of the recall process, a Toy-data problem was considered. First, we modeled with Toy-data to identify five abstract concepts. The proposed regulation algorithm utilized the learned intra-layer weight to determine the excitatory, neutral, and inhibitory impact induced by peer nodes on one another. Two types of cued recall experiments were performed using the unique geometric back-propagation algorithm: first, single-cue recall (SCR), simulation where the recall was simulated by activating only one abstract concept; second, multiple-cue recall (MCR) operation to retrieve the activation vector at the input level by injecting multiple cues at the abstract nodes. In SCR experiments, the regulation induced by peer nodes improved the recalled values. However, the observations with MCR operations were promising because they retrieved identical activation, as expected.

The benchmark MNIST dataset was used to exhibit cued recall as blends of learned abstract concepts. A two-layered model was generated with RAN to obtain thirty abstract concepts generically representing digits. In the multiple binary valued cue recall (MBVCR) experiment, multiple abstract nodes were injected with high activation to recall as blends of digits. Interestingly, it was observed in all the experiments that the blend of abstract nodes recalled an image of the digit that they represent at the abstract Level. The blend of different digits also produced some intriguing outcome; for exmaple, a blend of 2 and 5 recalled $x$, and a blend of 0 with 1 looked like a $\Phi$ symbol. The MCR operations were interesting as upon injecting the multiple cues, the recalled image was very similar to the expected recalled image.

Both the experiments displayed how oncepts can be contextually associated and impact each other's activation through regulation. Furthermore, with cue recall operations, it can be concluded that the more cues injected to an abstract concept, the better the obtained recall results. For future work, we intend to perform conceptual combination experiments and study the aspects of creative concept retrievals with the geometric back-propagation algorithm.

**Author Contributions:** R.S. performed state of the art, developed and implemented the methodology, carried out data selection and methodology validation, and prepared the original draft of the article. B.R. supervised the research work performed the formal analysis, review and editing, took care of funding. A.M.P. conceived the study plan and methodology, supervised the investigation, methodology development and implementation. A.C. supervised the research work, performed formal analysis, review and edition, managed funding. All authors have read and agreed to the published version of the manuscript.

**Funding:** The work presented in this paper was partially carried out in the scope of the SOCIALITE Project (PTDC/EEI-SCR/2072/2014), co-financed by COMPETE 2020, Portugal 2020—Operational Program for Competitiveness and Internationalization (POCI), European Union's ERDF (European Regional Development Fund), and the Portuguese Foundation for Science and Technology (FCT). This work was also partially funded by project ConCreTe. ConCreTe acknowledges the financial support of the Future and Emerging Technologies (FET) program within the Seventh Framework Programme for Research of the European Commission, under FET grant number 611733.

**Data Availability Statement:** In this work we used two datasets: first is the Toydata it is availabel at R.S. GitHub account at (https://github.com/rahulsharma-rs/datasets/blob/master/synthetic-data/Toy-dataRANstestclusterchapter6.csv accessed on 30 January 2021); second dataset is an image dataset named MNIST which is available in public domain (http://yann.lecun.com/exdb/mnist/ accessed on 30 January 2021).

**Acknowledgments:** I would like to express special thanks to Christine Zhang, Assistant Editor of the Applied Science Journal for her help in finding funds to support this article's publication cost.

**Conflicts of Interest:** The authors declare no conflict of interest.

## Abbreviations

The following abbreviations and Notations that are used in this manuscript:

| Abbreviations | Description |
| --- | --- |
| CAC | Convex Abstract Concept |
| CACC | Convex Abstract Concept Creation |
| CACUAP | Convex Abstract Concept Upward Activation Propagation |
| CCI | Convex Concept Identification |
| CCILL | Convex Concept Inter-Layer Learning |
| CCILW | Convex Concepts Inter-Layer Weights |
| CI | Concept Identifier |
| CRPD | Cluster Representative Data Points |
| CSRL | Concept Similarity Relation Learning |
| CSRW | Concept Similarity Relation Weights |
| E-A | Expected Activation |
| GBP | Geometric Back Propagation |
| IL | Intra-Layer |
| ILWs | Inter-Layer Weights |
| MBVCR | Multiple Binary Valued Cue Recall |
| MCR | Multiple Cue Recall |
| RAN | Regulated Activation Network |
| SCR | Single Cue Recall |
| **Notations** | **Description** |
| $W$ | Convex Concept Inter-layer weight matrix |
| $w$ | Similarity Relation weight matrix |
| $C$ | Cluster center or Centroids |
| $A$ | Output Activation |
| $a$ | Input Activation |
| $i, k, j$ | Variables to represent node index for 0th, 1st and 2nd *layer* respectively |
| $m, n$ | Arbitrary node indexes for any layer |
| $I$ | *I*th instance of input data |
| $f(x)$ | Transfer function to obtain similarity relation |
| $t$ | Variable used to depict intermediate index |
| $n_a$ | Size of input Vector at Layer-0 |
| $n_A$ | Size of Convex Abstract Concept vector at Layer-1 |

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
