# Peer review of "Emulating Cued Recall of Abstract Concepts via Regulated Activation Networks"

_applsci, doi:10.3390/app11052134_

Round 1
Reviewer 1 Report
This is a very interesting and well-written paper; there is a robust methodological approach that is clearly explained, and the evidence presented is extensive. One aspect that I would urge the authors to look into is the length. The paper is missing focus and clarity partly because of its length and much can be achieved through mainly editorial work to sharpen the content. I believe that will make the paper much clearer and shorter.
More specific observations: you are using the MNIST dataset for the demonstration; explain how potentially you take account of cross validation with other datasets.
Please provide a clear outline of your specific contributions in the introduction; these are not clearly mentioned.
In the Introduction, please provide the abstraction hierarchy in a diagrammatical representation
You also first introduce the recall procedure and need to clarify what it refers to (e.g. recall procedure in what process?)
IN Line 44 there is a grammar issue in “ pairs of Abstract Concepts are relevantly related to one another,” — ‘relevantly’ may need to change.
The intro provides a somewhat conflated review of relevant approaches and the motivation of the work, but none are clear. I would suggest shortening and crystallising the contribution (if need be, breaking it down into a numbered list).
The related work section is possibly too long. While the importance of the psychology work needs to be stressed, I think this section can be further reviewed for brevity.
Line 126 reads “in this work” and it is unclear whether you refer to your work or the one presented in [57]. You need to clarify. Finally (lines 151-157), while emanating from the above discussion, does not make a clear linkage to it.
It appears that section Background can be shorter in its detail - almost be absorbed by the previous one. (Note that all such recommendations for shortening pertain to legibility of the manuscript through maintaining focus). Sections 4 and 5 are very complete and clear, albeit on the long side. No specific recommendations are offered here, other that the general editorial suggestion provided above.
Finally, the conclusions need to become far clearer as to revisiting the contribution and providing the connection to how the contribution was achieved.
Author Response
We would like to take this opportunity to thank you for your time. We have carefully reviewed the comments and have revised the manuscript accordingly. We believe that the revised paper is a significant improvement over the manuscript. Specifically, we have:
(1) We have rewritten the sections of the Introduction.
(2) We have revised some sentences with grammatical mistakes.
(3) In the introduction, we supplemented the specific objectives and methods used in this paper.
(4) The content of related work has been reduced significantly.
(5) The Background section has been removed, and its content has been merged with the Section describing RAN’s modeling.
(6) The Conclusion is also improved.
Detailed Responses to the Reviewer 1’s comments :
This is a very interesting and well-written paper; there is a robust methodological approach that is clearly explained, and the evidence presented is extensive. One aspect that I would urge the authors to look into is the length. The paper is missing focus and clarity partly because of its length and much can be achieved through mainly editorial work to sharpen the content. I believe that will make the paper much clearer and shorter.
More specific observations: you are using the MNIST dataset for the demonstration; explain how potentially you take account of cross validation with other datasets.
Response: Yes, it is possible to use any kind of data provided that we can perform clustering operations on the data.
Please provide a clear outline of your specific contributions in the introduction; these are not clearly mentioned.
Response: At the end of paragraph 4 following content is added to highlight the contribution of the article: “To summarize, the following are the main contributions of the article: first, the impact factor calculation to determine the inhibitory, excitatory, or neutral effect of one node over other; second, the novel Intra-Layer Regulation algorithm to use impact factor to regulate the activation of other concepts; third, the novel Geometric Back-Propagation Algorithm; and Recall simulations using Geometric Back-Propagation Algorithm.”
In the Introduction, please provide the abstraction hierarchy in a diagrammatical representation
Response: A new Figure 1 has been added to the article to describe the hierarchical relationship among the abstract and concrete concepts, Figure 1 is also referenced in the First paragraph of the Introduction.
You also first introduce the recall procedure and need to clarify what it refers to (e.g. recall procedure in what process?)
Response: I have changed the first sentence of the second paragraph of Introduction to “The prime aspect of this article is to emulate the recall procedure, that can be viewed as the cognitive process of remembering”, to make the sentence more meaningful with respect to cognitive processes.
IN Line 44 there is a grammar issue in “ pairs of Abstract Concepts are relevantly related to one another,” — ‘relevantly’ may need to change.
Response: The second paragraph has been re-written completely and the above-mentioned issue is also addressed while re-phrasing the paragraph.
The intro provides a somewhat conflated review of relevant approaches and the motivation of the work, but none are clear. I would suggest shortening and crystallising the contribution (if need be, breaking it down into a numbered list).
Response: Paragraph 3 of the introduction is completely re-written according to the recommendations of the reviewer
The related work section is possibly too long. While the importance of the psychology work needs to be stressed, I think this section can be further reviewed for brevity.
Response: Last three paragraphs in the section of related work are replaced by one paragraph describing the connection of the article with the related work.
Line 126 reads “in this work” and it is unclear whether you refer to your work or the one presented in [57]. You need to clarify. Finally (lines 151-157), while emanating from the above discussion, does not make a clear linkage to it.
Response: The ambiguity due to the phrase “in this work” is resolved by replacing it with a new phrase “In this article” to indicate that I am referring to the current article. Last three paragraphs in the section of related work are replaced by one paragraph describing the connection of the article with the related work.
It appears that section Background can be shorter in its detail - almost be absorbed by the previous one. (Note that all such recommendations for shortening pertain to legibility of the manuscript through maintaining focus). Sections 4 and 5 are very complete and clear, albeit on the long side. No specific recommendations are offered here, other that the general editorial suggestion provided above.
Response: The background section is removed from the article and the biological inspiration of RAN’s regulation operation is merged with section 3 in the beginning.
Finally, the conclusions need to become far clearer as to revisiting the contribution and providing the connection to how the contribution was achieved.
Response: The content of the conclusion has been revised to make it more meaningful with respect to the content of the article. Following is the content added to the beginning of the conclusion “Recall is a cognitive process that can also be seen as an act of remembering a concept. Concepts are normally perceived in a hierarchical form where the Concrete Concepts occupy the lower level and the Abstract Concepts take up the relatively higher level in the hierarchy. According to context availability theory, the context among the Concrete Concepts is easily determined when compared to Abstract Concepts, hence, their comprehension and recall are also difficult. However, if we can relate abstract concepts to one another it is possible to deduce a contextual relationship among them. In this article, we exploit the intra-layer associations learned among the concepts (including abstract concepts) using RAN's modeling to establish context among the concepts. We use this context-related information to induce regulatory effect among the concepts and further to simulate the recall operations.”

Reviewer 2 Report
Although it is not my field of study, I apply neural network techniques and artificial intelligence with other types of data. However, I have enjoyed reading your article, I believe that this study may have an impact among readers.I would like to propose some small modifications;
- The indentation on the left is very large, I understand that it is because of the notes that are in 2. Could it be put in some other way? I think the article could be shorter with this modification.
- I think figures 4 and 6 could be a bit bigger because they are relevant. On the other hand, the numbers in figure 5 could be larger, you could evaluate expressing them in scientific notation.
- Finally, there are some considerable blank spaces before and after some figures (7, 8) and tables (2,3,4,5,7).
Author Response
We would like to take this opportunity to thank you for your time. We have carefully reviewed the comments and have revised the manuscript accordingly. We believe that the revised paper is a significant improvement over the manuscript. Specifically, we have:
(1) We have proof read the menuscript and tried to improve the language of the article.
(2) We have made all structure related changes suggested by the reviewer.
Detailed Responses to Reviewer 2’s comments:
Although it is not my field of study, I apply neural network techniques and artificial intelligence with other types of data. However, I have enjoyed reading your article, I believe that this study may have an impact among readers.I would like to propose some small modifications;
- The indentation on the left is very large, I understand that it is because of the notes that are in 2. Could it be put in some other way? I think the article could be shorter with this modification.
Response: I agree. I will ask the editor about the possibility to make this change because I am using a template provided by the journal.
- I think figures 4 and 6 could be a bit bigger because they are relevant. On the other hand, the numbers in figure 5 could be larger, you could evaluate expressing them in scientific notation.
Response: The size of Figures has been increased.
- Finally, there are some considerable blank spaces before and after some figures (7, 8) and tables (2,3,4,5,7).
Response: All space problems related to the figures have been addressed and they have been resized according to the page size.

Reviewer 3 Report
- "These synapses occur when the axon of a neuron connects to the axon of another neuron instead of to its dendrites". When you read this sentence and try to understand figures 1a and 1b it is confusing. The axon representation on both figures should be similar.
- The author has mention everywhere the use of Toy-data but they have failed to mention the basic information related to the toy-data for the reader.
- The conclusion looks lengthy, it should be modified.
- As mention in ref [9], the author had previously published a similar concept. What advancement you have done relative to your previous publication.
Author Response
We would like to take this opportunity to thank you for your time. We have carefully reviewed the comments and have revised the manuscript accordingly. We believe that the revised paper is a significant improvement over the manuscript. Specifically, we have:
(1) Addressed the All the issues suggested by the reviewer.
(2) The Conclusion is also improved.
Detailed Responses to Reviewer 3’s comments :
- "These synapses occur when the axon of a neuron connects to the axon of another neuron instead of to its dendrites". When you read this sentence and try to understand figures 1a and 1b it is confusing. The axon representation on both figures should be similar.
Response: We have redrawn Figure 1b and removed Figure 1a to clear the ambiguity. The new figure is Figure 2.
- The author has mention everywhere the use of Toy-data but they have failed to mention the basic information related to the toy-data for the reader.
Response: Description about the Toy-day has been added to section 3.2. The following content has been added “The Toy-data is synthetically produced by generating a 2-dimensional dataset with five classes. In Figure 3 we can see that out of the five clusters three are far apart from one another, however, two clusters are very close to each other. This arrangement of clusters was introduced into the Toy-data problem to demonstrate the Excitatory and Inhibitory impact of Concepts, representing each cluster at an Abstract level. The dataset consists of $1800$ data instances with an equal distribution of all the classes.”
- The conclusion looks lengthy, it should be modified.
Response: The content of the conclusion has been revised to make it more meaningful with respect to the content of the article. Following is the content added to the beginning of the conclusion “Recall is a cognitive process that can also be seen as an act of remembering a concept. Concepts are normally perceived in a hierarchical form where the Concrete Concepts occupy the lower level and the Abstract Concepts take up the relatively higher level in the hierarchy. According to context availability theory, the context among the Concrete Concepts is easily determined when compared to Abstract Concepts, hence, their comprehension and recall are also difficult. However, if we can relate abstract concepts to one another it is possible to deduce a contextual relationship among them. In this article, we exploit the intra-layer associations learned among the concepts (including abstract concepts) using RAN's modeling to establish context among the concepts. We use this context-related information to induce regulatory effect among the concepts and further to simulate the recall operations.”
- As mention in ref [9], the author had previously published a similar concept. What advancement you have done relative to your previous publication.
Response: The citation [9] is the conference article which was extended to the article cited by [11], In this article, we used the RAN’s method described in these articles to build the models for Toy-data and MNIST data. We have revised the 4th paragraph of the introduction to highlight the main contributions of the article. Following is the content added to the introduction “To summarize, the following are the main contributions of the article: first, the impact factor calculation to determine the inhibitory, excitatory, or neutral effect of one node over other; second, the novel Intra-Layer Regulation algorithm to use impact factor to regulate the activation of other concepts; third, the novel Geometric Back-Propagation Algorithm; and Recall simulations using Geometric Back-Propagation Algorithm.”

Reviewer 4 Report
In this paper, the authors presented their research on abstract concept learning via regulated activation network. A geometric back-propagation mechanism is used to simulate the recall of learned abstract concepts. The experimental results confirmed the effectiveness of their design and hypothesis. This research is an extension of the authors' prior work which was appropriately cited to highlight the new findings. Overall, the paper is well-written and the research quality is solid.
Minor issues exist in wording and punctuation (e.g., lines 258-259, "Since, ..."). Proofreading should be done before publication.
Author Response
We would like to take this opportunity to thank you for your time and positive comments. We have carefully reviewed the comments and have revised the manuscript accordingly. We believe that the revised paper is a significant improvement over the manuscript. Specifically, we have:
Detailed Responses to Reviewer 4’s comments :
In this paper, the authors presented their research on abstract concept learning via regulated activation network. A geometric back-propagation mechanism is used to simulate the recall of learned abstract concepts. The experimental results confirmed the effectiveness of their design and hypothesis. This research is an extension of the authors' prior work which was appropriately cited to highlight the new findings. Overall, the paper is well-written and the research quality is solid.
Minor issues exist in wording and punctuation (e.g., lines 258-259, "Since, ..."). Proofreading should be done before publication.
Response: The punctuation issue and the similar grammatical and punctuation issues have been addressed by reviewing the entire manuscript for language-rated errors.
